



# 1 From Tsunami Risk Assessment to Disaster Risk Reduction. The case
# 2 of Oman

Ignacio Aguirre Ayerbe[1], Jara Martínez Sánchez[1], Íñigo Aniel-Quiroga[1], Pino González-Riancho[2], María
Merino[1], Sultan Al-Yahyai [3], Mauricio González[1], Raúl Medina[1]
[1]Environmental Hydraulics Institute "IHCantabria", University of Cantabria, Santander, 39011, Spain
[2]GFA Consulting Group, Hamburg, 22359, Germany
[3]Directorate General of Meteorology and Air Navigation. Public Authority for Civil Aviation, Muscat, 111, Oman
*Correspondence to*: I. Aguirre Ayerbe (ignacio.aguirre@unican.es)
**Abstract.** Oman is located in an area of high seismicity, facing the Makran Subduction Zone, which is the major source of
earthquakes in the eastern border of the Arabian plate. These earthquakes, as evidenced by several past events, may trigger a
tsunami event. The aim of this work is to minimize the consequences that tsunami events may cause in coastal communities
by integrating tsunami risk assessment and risk reduction measures as part of the risk-management preparedness strategy. An
integrated risk assessment approach and the analysis of site-specific conditions permitted to propose target-oriented risk
reduction measures. The process included a participatory approach, involving a panel of local and international experts.
One of the main concerns of this work was to obtain a really useful outcome for the actual improvement of tsunami risk
management in Oman. This goal was achieved through the development of comprehensive and functional management tools
such as the Tsunami Hazard, Vulnerability and Risk Atlas and the Risk Reduction Measures Handbook, which will help to
design and plan a roadmap towards risk reduction.

## 19 1 Introduction

Tsunamis are low-frequency natural events but have a great destructive power when striking coasts around the world, involving
loss of life and extensive damage to infrastructures and coastal communities worldwide. Between 1996 and 2015, estimated
tsunami disaster losses reached 250,000 lives, more than 3,500,000 affected people and more than 220,000 million of USD
(International Disaster Database, EM-DAT; UNISDR/CRED, 2016).
Oman is located in an area of high seismicity, facing the Makran Subduction Zone (MSZ), which is the major source of
earthquakes in the eastern border of the Arabian plate (Al-Shaqsi, 2012). These earthquakes may trigger a tsunami event, as
evidenced at least three times in the past (Heidarzadeh et al., 2008; Jordan, 2008). The high potential for tsunami generation
of MSZ makes it one of the most tsunamigenic areas of the Indian Ocean. The most recent tsunami event of seismic origin
was the 1945 Makran tsunami, which caused more than 4,000 fatalities and property losses in Iran, Pakistan, Oman and the
United Arab Emirates (Heidarzadeh et al., 2008, Mokhtari, 2011). Similar episodes may occur again in this area.
In addition to the tsunami threat on the coast of Oman, the rapid development and industrialization of this area explains the
need to develop specific studies on tsunami vulnerability and risk, especially in the northern low-lying coastal plain, which is
the most densely populated and the most exposed to the MSZ.
Suitable tsunami vulnerability and risk assessments are essential for the identification of the exposed areas and the most
vulnerable communities and elements. They allow identifying appropriate site-specific risk management strategies and
measures, thus enabling to mainstream disaster risk reduction (DRR) into development policies, plans and programs at all
levels including prevention, mitigation, preparedness, and vulnerability reduction, considering its root causes.
Most methods for risk assessment are quantitative or semi quantitative (usually indicator-based). Quantitative risk assessments
are generally better related to the analysis of specific impacts, which require large scales and high resolution for all the
components composing the risk. Results are usually expressed in terms of potential losses both economic (derived from



building damage or even infrastructure damage) and human (derived from mortality estimations). There are several works
following this approach, among others Løvholt et al., 2014, Suppasri et al. in 2011 and 2013, Tinti et al. (2011) and Valencia
et al. (2011) within the frame of the European project SCHEMA[1], Leone et al. (2011), Mas et al. (2012), Berryman et al.
(2005), Sugimoto et al. (2003), Sato et al. (2003), Koshimura et al. (2006), Jonkman et al. (2008), and Harbitz et al. (2016).
Although not as common, quantitative risk assessments are sometimes applied at global scale such as the case of the GRM -
Global Risk Model (last version in GAR, 2017), which addresses a probabilistic risk model at a world scale to assess economic
losses based on buildings damage (Cardona et al., 2015).
However, when the scope requires a holistic and integrated approach in which several dimensions, criteria and variables with
different magnitudes and ranges of values are to take into consideration, such as the case of the present work, it is necessary
to apply an indicator-based method. Some works following this approach may be found in ESPON (2006), Dall'Osso et al.
(2009a, 2009b), Taubenböck et al. (2008), Jelínek (2009, 2012), Birkmann et al. (2010, 2013), Strunz et al. (2011), Aguirre-
Ayerbe (2011), Wegscheider, et al. (2011), González-Riancho et al. (2014), the European TRANSFER[2] project, the Coasts at
Risk report (2014), the INFORM Global Risk Index (INFORM, 2017) and The World Risk Report (last version: Garschagen
et al., 2016).
Nevertheless, very few of them tackle with the direct link between integrated tsunami risk results and risk reduction measures
(RRM). González-Riancho et al., (2014) propose a translation of risk results into disaster risk management options and
Suppasri et al. (2017) describe some recommendations based on the lessons learned in recent tsunamis.
Therefore, it has been identified that there is not a clear applicability of science-based tsunami hazard and vulnerability tools
to improve actual DRR efforts, highlighting a general disconnection between technical and scientific studies and risk
management.
This work attempts to be complementary to preceding efforts and to fill the gap found in previous studies. The developed
methodology is based on the direct relationship found between risk components (hazard, exposure and vulnerability) and
specific DRR measures and integrates tsunami risk assessment and site-specific characteristics to select a suitable set of
tsunami countermeasures. The ultimate goal is the application of the method and the generation of useful management tools
to minimize the consequences that a potential tsunami could have on the coast of Oman.
**2 Methodology**
The methodology comprises two main phases: (i) the integrated tsunami risk assessment and (ii) the identification, selection
and prioritization of appropriate DRR measures. These two different but complementary tasks will guide the entire
methodology applied in this work.
As regards the conceptual framework, the methodology applied is fundamentally adapted from the definitions of UNISDR
(2004, 2009), ISO/IEC Guide 73 (2009), UNESCO (2009b) and UN (2016). Accordingly, the sequence of the work is
summarized schematically in Figure 1. Within the disaster risk assessment phase and prior to any risk study, it is necessary to
define the consequence to be analysed and the type of result pursued (for example, the estimation of buildings damages or, the
community's affection from a holistic perspective, as the case presented in this article). The establishment of this main goal
determines the specific method, the dimensions to include in the study and the spatial and temporal scales (point 1 of Figure

75 1).


---

[1] SCHEMA Project: Scenarios for Hazard-induced Emergencies Management. European 6th Framework Programme Project no. 030963, August 2007 - October 2010.

[2] TRANSFER project: Tsunami Risk and Strategies for the European Region. European 6th Framework Programme no. 37058, October 2006-September 2009.



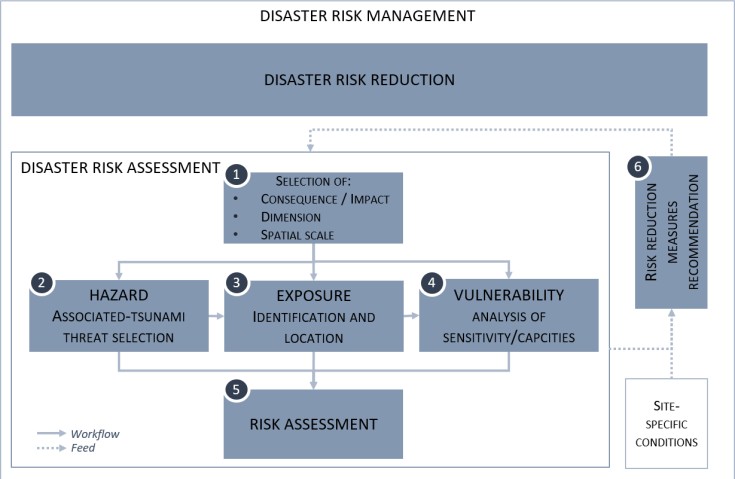


**Figure 1. Schematic workflow**
Next, the assessment of the hazard, explained in detail in section 2.1 Hazard Assessment, requires the selection of the variable
associated to the event (e.g. inundation depth) mainly determined by the general goal defined in the first step. The hazard
evaluation drives to the analysis of the individuals and elements exposed (e.g. people, buildings and infrastructures located in
a flooded populated area) together with its vulnerability (e.g. sensitive age groups). The risk assessment is performed by the
combination of the vulnerability assessment -of what is exposed- and the hazard intensity (points 3, 4 and 5 of Figure 1,
explained in detail in sections 2.2 Vulnerability assessment and 2.3 Risk Assessment). Both, exposure, vulnerability and the
integration of all risk components, circumscribed to a given spatial, cultural and socioeconomic context, are necessary for the
preliminary selection of risk reduction strategies and measures. These countermeasures are essential to prevent new and reduce
existing risk, as stated by UN (2016), contributing to the strengthening of resilience and reduction of disaster losses (point 6
in Figure 1. Schematic workflow, detailed in section 2.4 Risk reduction measures).
The determination of the efficiency of each proposed countermeasure is essential for the success of the risk reduction planning.
When an appropriate countermeasure is selected, the overall risk assessment must be conducted again to understand how and
to what extent it will actually reduce the risk.
DRR measures are framed in the disaster risk management cycle proposed below, which brings together four main strategies
for risk reduction (Figure 2): (i) prevention and (ii) preparedness strategies in the pre-event stage and (iii) emergency/response
and (iv) recovery in the post-event phase. Each of the strategies includes several actions that may be overlapped on time and
that may even belong to more than one strategy. At the centre of the figure, research is presented as an essential element to
improve disaster management enriching the process through the integration of various disciplines and studies. This particular
study focuses on the strategies related to the pre-event phase: the prevention and the preparedness, which are explained in
section 2.4 Risk reduction measures.





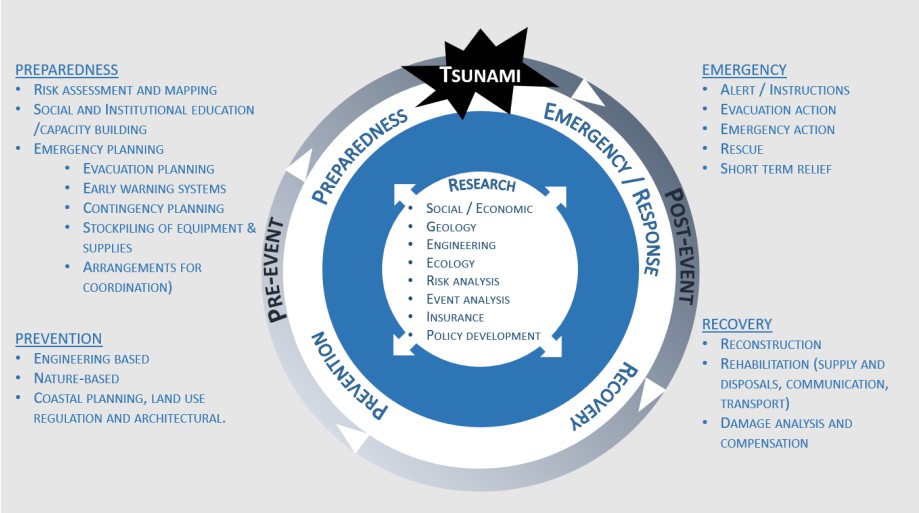


**Figure 2. Disaster risk management cycle.**

Risk and vulnerability assessments are performed both for a specific place and at a specific time. For this reason, both the
analysis and the proposal of measures for risk reduction must be updated periodically, considering the changes that may occur
over time and their influence on the results, such as a significant variation in population, land-use changes, new constructions
or new lessons learnt.
The involvement of key local stakeholders and decision-makers in coastal risk management is essential throughout the entire
process, both to include their knowledge and expertise and to enhance the usefulness of the results of the project throughout
their encouragement. Thus, a stakeholder panel composed of local and international experts on coastal risks and risk
management supported the entire process, driven to actively participate and collaborate to achieve the goal of DRR. Their
main contribution focused on the validation of the methodological approach, the identification of hot spots and the analysis of
the technical, institutional and financial capacities of the country for implementing each one of the countermeasures. In the
last stage of the study, they prioritized each measure according to their knowledge and expertise.

## 2.1 Hazard Assessment

The hazard analysis allows determining the areas that would be affected due to the potential tsunamis that may strike the study
area. The analysis is carried out considering the worst possible tsunami scenarios based on the seism-tectonic characterization
of the area, so that the maximum impact that a tsunami would cause is calculated. Similar approaches may be found in Jelínek
et al. (2009, 2012), Álvarez-Gómez et al. (2013) and Wijetunge (2014) among others. The deterministic tsunami hazard
analysis allows identifying, locating and analysing the elements at risk in a conservative approach. It is worth considering this
method when dealing with intensive risks, i.e. derived from low frequency but high severity hazards, such as tsunamis, where
the catastrophic consequences of the impact are complex and difficult to estimate.
The analysis considered potential earthquake sources since it is the most common tsunami generation mechanism in the area,
where the Arabian plate (moving northwards) and the Eurasian plates converge. A seism-tectonic analysis was performed to
identify and characterise the major seismic structures with capacity to generate a tsunami affecting the coast of Oman (see
Aniel-Quiroga et al., 2015). The study area was divided in three tectonically homogeneous zones including eleven main
structures. The geometrical characterization of the fault planes (from the tectonics and the focal mechanisms analysis) allowed
identifying 3181 focal mechanisms with a magnitude varying from Mw 6.5 and Mw 9.25.
Once these scenarios are established, the analysis includes the characterization of the quake (fault location, magnitude of the
quake, length and width of the fault, fault dislocation angles, epicentre location and focal depth of the epicentre) and the sea

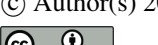



level. The numerical modelling applied to conduct the simulations is COMCOT (Wang, 2009), which is a shallow water
equation model that uses Okada model to generate the initial deformation of the sea surface. Based on the bathymetry, the
propagation of each potential tsunami is modelled from the source to the coast. Finally, according to the topography, the coastal
area is flooded, with a final resolution of 40 m onshore.

The approach is described in detail in Aniel-Quiroga et al. (2015) and is based on the works of Álvarez-Gómez et al. (2014)
and Gutiérrez et al. (2014).
Figure 3 shows the distribution of the major seismic structures and the number of events propagated for each of them. The
seism-tectonic study was particularly focused in the Makran subduction zone, since it is possibly the most active area in the
western Indian Ocean and located very near the north coast of Oman.

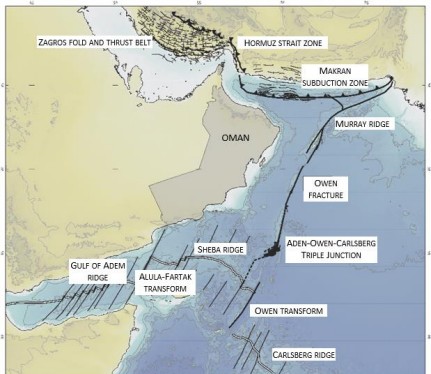

| | Tectonic Structure/Zone | Events propagated |
|---|---|---|
| 1 | Gulf of Adem Ridge | 44 |
| 2 | Alula Fartak | 1 |
| 3 | Aden-Owen-Carlsberg Triple Junction | 14 |
| 4 | Sheba Ridge | 65 |
| 5 | Owen Transform | 3 |
| 6 | Carlsberg Ridge | 74 |
| 7 | Owen Fracture | 16 |
| 8 | Murray Ridge | 33 |
| 9 | Homuz Strait Zone | 48 |
| 10 | Zagros Zone | 148 |
| 11 | Makran Subduction Zone | 2735 |

**Figure 3. Main seismic areas surrounding the study area and number of events propagated for each area**
On one side, the complete set of the 3181 scenarios were included in a tsunami-scenarios database, which is the basis for the
establishment of the early warning system in the country. On the other, seven scenarios were selected to perform the
deterministic hazard assessment, including the historical event of 1945, which took place in the Makran subduction zone
(Heidarzadeh et al., 2008). These scenarios were aggregated into a map that shows at each point of the study area the worst
possible situation. This enveloping map is the base for the risk assessment and includes the variables of inundation depth,
water velocity and drag level (multiplication of water velocity times inundation depth).
Hazard variables were finally classified into five intensity levels, which are described in section 2.3 Risk Assessment. Tsunami-
drag variable classification is based on previous works carried out by Xia et al. (2014) Jonkman et al. (2008), Karvonen et al.
(2000), Abt et al. (1989), which establish different thresholds related to the people stability. As for the inundation depth
variable, the classification is based on the work developed in the SCHEMA project (Tinti et al., 2011) to establish building
damage levels, based on empirical damage functions considering building materials and water depth.

## 2.2 Vulnerability assessment

The method applied to assess the vulnerability relies on an indicator-based approach. The process include three main stages:
(a) the definition of criteria for selecting the dimensions and variables to be analysed for the exposed elements, (b)
establishment, calculation and classification of indicators and (c) the construction of vulnerability indexes and its classification.
These steps are explained in the following paragraphs.
Two different dimensions are selected: human and infrastructures, with the aim of developing an analysis with a human-centred
perspective. On one side, the human dimension allows analysing the intrinsic characteristics of the population. On the other,
the infrastructure dimension allows the analysis of buildings and critical facilities, to consider their potential worsening
implications for the populations, following the rational described in González-Riancho et al. (2014). In this sense, it is




considered that an increase in the number of victims is likely to occur due to the loss or damage of emergency services, or the
recovery capacity may decrease due to the loss of strategic socioeconomic infrastructures such as ports.
The criteria to analyse the human dimension are, the population capacities related to their mobility and evacuation speed, and
the ability to understand a warning message and an alert situation. The criteria determined to analyse the infrastructure
dimension are, the critical buildings housing a large number of people (schools, hospitals, etc.), the emergency facilities and
infrastructures, the supply of basic needs, the building and infrastructures that could generate negative cascading effects, and
the economic consequences.
Consequently, a set of 11 indicators has been defined (see Table 1) to develop a framework that allows to encompass the major
issues related to the community's vulnerability This framework was developed in agreement with local stakeholders and
international experts through the participatory process.

| Index | | Indicator | Variable |
|---|---|---|---|
| Human Vulnerability Index | Human Exposure | H1 - Population | Number of persons exposed |
| | Human Sensitivity | H2 - Sensitive age groups | Number of persons <10 and > 65years |
| | | H3 - Disability | Number of disabled persons (physical / intellectual) |
| | | H4 - Illiteracy | Number of illiterate persons |
| | | H5 - Expatriates | Number of expatriates |
| Infrastructure Vulnerability Index | Infrastructures Exposure | I1 - Buildings and infrastructures | Number of exposed buildings and infrastructures |
| | Infrastructures Sensitivity | I2 - Critical buildings | Number of critical buildings (health, educational, religious, cultural, governmental) |
| | | I3 - Emergency | Number of emergency infrastructures (civil defence, police, firemen, military, royal guard) |
| | | I4 - Supply | Number of water supply (desalination plants) and energy supply (power plants) infrastructures |
| | | I5 - Dangerous | Number of dangerous/hazardous infrastructures |
| | | I6 - Strategic | Number of strategic infrastructures (ports and airports) |

**Table 1. Exposure and sensitivity indicators built for the tsunami vulnerability assessment in Oman.**
Indicators H1 and I1 identifying and locating the number and type of exposed population and infrastructures respectively.
The human indicators H2-H5 are oriented to measure weaknesses in terms of evacuation and reaction capacities of the exposed
population. Specifically, H2 and H3 are related to problems with mobility and evacuation velocity whereas H2, H3, H4 and
H5 are related to difficulties in understanding a warning message and an alert situation.
The infrastructure indicators I2-I6 measure the number of critical facilities and buildings that would be affected by
administrative area, bearing in mind the implications for the population. I2 provides the number of buildings that would require
a coordinated and previously planned evacuation due to the high number of people in them (in some cases sensitive population),
such as hospitals, schools, geriatrics, malls, stadiums, mosques, churches, etc. I3 calculates the loss of emergency services that
are essential during the event. I4 reports on the potential number of power plants and desalination plants affected, hindering
the long-term supply of electricity and water to local communities. I5 analyses the generation of cascading impacts that could
take place due to affected hazardous/dangerous industries. Finally, I6 considers the loss of strategic ports and/or airport
infrastructures, essential for the economy of the country and the local livelihoods (fishing ports).
The construction of vulnerability indexes is performed through the weighted aggregation of the previously normalized
indicators (via the min-max method). Aggregated indexes are then classified considering the data distribution via the natural
breaks method (Jenks, 1967) and grouped in five classes, obtaining homogeneous vulnerability areas that are expected to need
similar DRR measures.


Indicators and indexes have been applied to every wilayat along the coast of Oman (wilayat is an administrative division in
Oman). Comparable results are obtained among all areas due to the methods of normalization and classification, which take
into account the values of the index for all areas when establishing classes' thresholds. This method depends on the distribution
of the data, therefore the study of any index evolution over time, for comparable purposes, must maintain the thresholds
established in the initial analysis. In the same way, if new study areas were added, they should be included and new thresholds
should be established.
**2.3 Risk Assessment**
Risk results are obtained by combining hazard and vulnerability components through a risk matrix (Greiving et al., 2006;
Jelínek et al., 2009; Aguirre-Ayerbe, 2011; González-Riancho et al., 2014; Schmidt-Thomé, 2006; ESPON, 2006; IH
Cantabria-MARN, 2010 and 2012 projects). Classes derived from the hazard assessment are blended with vulnerability classes
as shown in Figure 4 to obtain two types of results, partial risks for each dimension and a combined risk result from the
weighted aggregation of both dimensions. The results are finally classified into five risk classes.

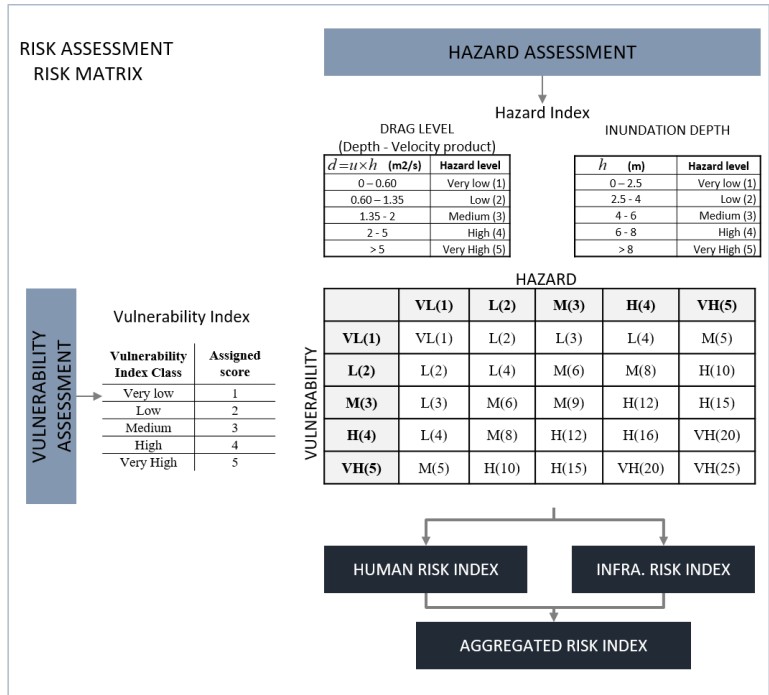

**Figure 4. Risk matrix combining hazard and vulnerability classes.**
The hazard variable differs according to each dimension of the study to analyse specifically the potential impacts. The drag
level variable (understood as the hazard degree for human instability based on incipient water velocity and depth) is applied
to the human dimension whereas the inundation depth variable is applied to the infrastructure dimension
The results obtained from the risk matrix reveal areas at high risk, which are expected to have serious negative consequences
due to the combination of hazard and vulnerability conditions. In-depth analysis of these areas allows to identify the causes of
these results and to propose adequate RRM according to each of the components, dimensions and variables considered to
perform the risk assessment.





**2.4 Risk reduction measures**

It has been developed a method to identify, recommend and prioritize a set of most-suitable alternatives for tsunami risk reduction based on the risk analysis and site-specific conditions. The very first step has been the development of a RRM catalogue, to finally obtain a set of site-specific and target-oriented countermeasures. This method facilitates the decision-making process by connecting scientific and technical results with risk management.

The work focuses on the straightforward feeding/reduction relation among the different risk components and the risk reduction measures focused on the pre-event stage (see Figure 5).

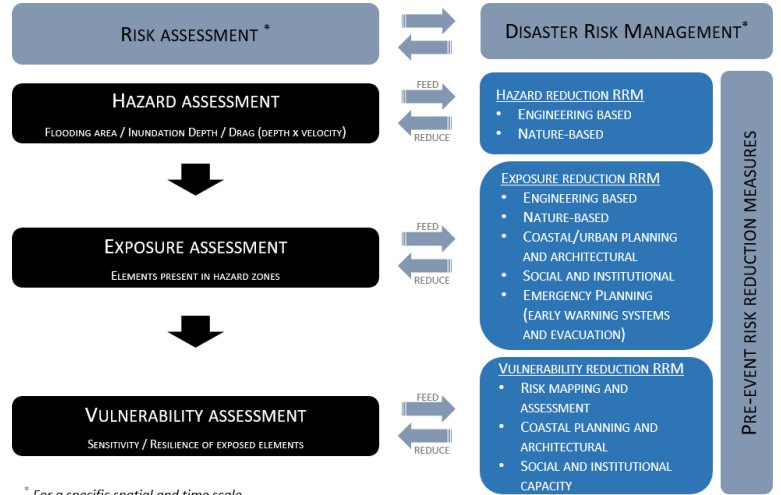

*For a specific spatial and time scale.

**Figure 5. Interactions between the different components of risk assessment and the pre-event approaches of risk reduction measures**

Accordingly, two main strategies are identified to achieve a long-term coastal flooding risk reduction: preparedness and prevention, which are based on the concepts defined by UN (2016) and UNISDR (2009).

Preparedness actions are focused on the knowledge, capacities and skills developed to anticipate and respond to the impacts of the event, and include the following: (i) risk assessment and mapping, (ii) social and institutional awareness, educational and capacity building measures, and (iii) emergency measures. The risk assessment and planning is the first step of the risk management cycle, providing essential guidance within the decision-making process. The social and institutional measures enhance the knowledge and capacities developed by communities and individuals to effectively anticipate and respond to the impacts of likely, imminent or current hazard events, as stated by UN (2016). The emergency measures ensure public safety by issuing alerts and planning evacuation of people and certain goods (e.g. vessels) at risk, to safe areas or shelters when a tsunami is detected. There are some other preparedness measures, which are oriented to the post-event phase of the disaster management, such as contingency planning, stockpiling of equipment and supplies and arrangement for coordination.

Prevention refers to actions that aim at shielding or protecting from the hazard through activities taken in advance, by reducing the hazard itself, the exposure to that hazard or the vulnerability of the exposed people or goods. These include (i) engineering-based measures, (ii) nature-based measures, and (iii) coastal planning and architectural measures. The engineering-based measures, i.e., controlled disruption of natural processes by using long term man-made structures (hard engineering solution) help to reduce the intensity of the hazard. The nature-based measures, i.e., the use of ecological principles and practices (soft engineering solution) help to reduce the intensity of the hazard and to enhance coastal areas safety while boosting ecological wealth, improving aesthetics, and saving money. The coastal planning and architectural measures, i.e. regulations and good practices, reduce the exposure and vulnerability mainly related to the infrastructure dimension.

Table 2 shows the set of RRM developed (based on UNFCC, 1999; Nicholls et al., 2007; UNESCO, 2009a, Linham et al., 2010), organised by strategies, approaches and specific goals.





| Strategy | Approach | Code | Mitigation measure | Specific Goal |
|---|---|---|---|---|
| Preparedness | Risk Mapping and Assessment | RA. 1 | Hazard, Vulnerability and Risk | V |
| | Social and institutional capacity | PR. 1 | Raising awareness | $E_t$ and V |
| | | PR. 2 | Capacity building | |
| | | PR. 3 | Education | |
| | Emergency planning | EM. 1 | Early Warning Systems | $E_t$ |
| | | EM. 2 | Evacuation planning | |
| Prevention | Engineering-based | EN. 1 | Seawalls and sea dykes | H |
| | | EN. 2 | Breakwaters | |
| | | EN. 3 | Movable barriers and closure dams | |
| | | EN. 4 | Land claim | |
| | Nature-based | NA. 1 | Managed realignment | H |
| | | NA. 2 | Beach nourishment | |
| | | NA. 3 | Artificial sand dunes and dune restoration | |
| | | NA. 4 | Living shorelines | |
| | | NA. 5 | Wetland restoration | |
| | Coastal Planning and Architectural | PL. 1 | Building standards | V |
| | | PL. 2 | Flood proofing | |
| | | PL. 3 | Coastal setbacks | $E_p$ |

**Table 2. Strategies, approaches, measures and specific goals for risk reduction derived from coastal risk due to tsunami hazard (H: hazard, $E_P$: permanent exposure, $E_t$: temporary exposure, V: vulnerability).**

The catalogue has been developed following this concepts and structure. Each measure is analysed and characterised by means of individual RRM-cards that include the specific objective pursued and description of the measure in several sections: rationale, preliminary requirements, supplementary measures, efficiency, durability and initial cost analysis. Each card includes a list of stakeholders involved in the implementation of the specific RRM in Oman, and the estimation of the current capacity for implementation, based on the information provided by the experts' panel. Each card also contains a scheme, several figures and a suitability analysis, which is performed through a SWOT analysis. Finally, it is incorporated a specific bibliographic reference list that permits a deeper study of each measure.

This RRM catalogue is the basis for the next step, the selection and prioritization of the specific set of countermeasures for each area. It is also worth to mention that a combination of measures from different approaches often offers an effective risk reduction strategy, even enhancing the performance of the individual measures when implemented at the same time.

### 2.4.1 Risk reduction measures selection and prioritization

The methodology for the selection and prioritization of the RRM has been designed to ensure its adequacy to site-specific conditions at local scale among those proposed in the catalogue. It is summarize in three main steps: (i) determination of the management units, (ii) selection of the recommended RRM through a decision matrix and (iii) the prioritization of RRM (see Figure 6).





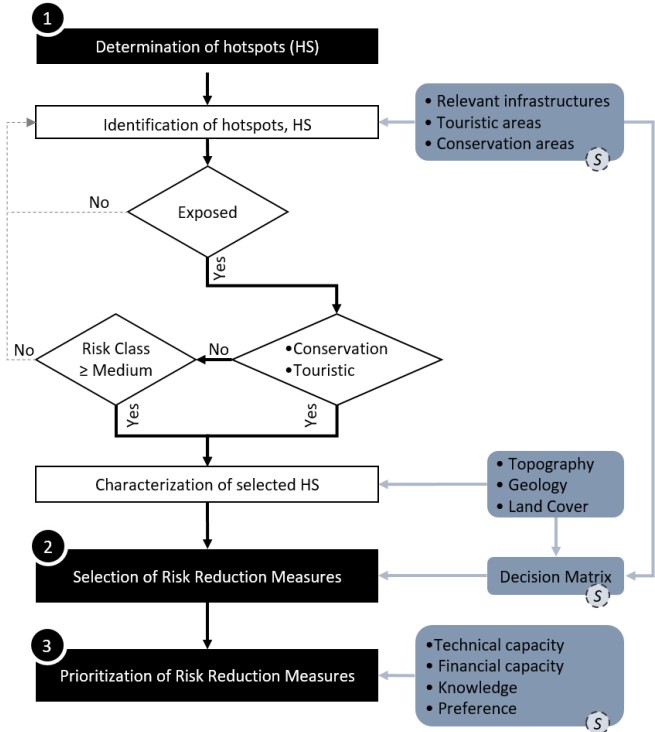

256

**Figure 6. Scheme of the methodology for the prioritization of recommended tsunami risk reduction measures (S: panel of local and international experts on coastal risk management and coastal stakeholder's participation).**

The first step is the determination of hotspots, which are the zones in which RRM will be further proposed. Coastal hotspots (HS) are identified in consensus with the stakeholder panel, including built-up populated areas and the following areas of special interest: (i) relevant infrastructures such as transport and communications infrastructures (airports and sea-ports), supply infrastructures (power and water) and dangerous infrastructures (refineries, dangerous industries areas and military bases); (ii) touristic regions, where there is significant seasonal variation in the population and (iii) environmental conservation areas, to consider the fragile and complex systems where the coastal ecosystems converge with the marine dynamics and the human activities, which include lagoons, mangroves and turtle nesting areas.

After the identification of the HS, it is evaluated whether they are exposed to tsunami hazard and if they exceed the risk class threshold as shown in Figure 6, in order to determine the units that will feed the decision matrix into the second phase. Because of their significance, the scarcity of data when performing the vulnerability assessment and the relevance given by local stakeholders, touristic regions and environmental conservation areas will move to the next step if the HS is exposed, regardless the risk level. In all other cases, for those HS under very low, low risk or not expose, no countermeasures will be assigned. The HS characterization is carried out by assigning elevation characteristics (highlighting low-lying areas and wadis), a geology categorization (bare consolidated or non-consolidated substratum) and the land cover (cropland, built-up areas and vegetation-covered areas).

The second stage consists in the preliminary assignment of RRM to each HS according to the decision matrix. The matrix, which was validated by the stakeholder panel, is fed by the specific characteristics of each HS and by type of HS, as described previously. Table 3 shows the decision matrix, already sorted by the ratings of key stakeholder on coastal risk management in Oman, as explained below.

The assignment of each recommended measure (highly recommended, recommended or not recommended) depends on the characteristics that have determined the type HS. On the one hand, the topography of the area, with a focus on the low-lying

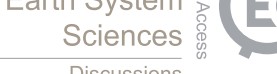



areas and wadis, where flooding occurs on a regular basis, at least annually. Likewise, the geology and land cover is analysed
to consider the bedrock and type of land use, that condition the suitability of one or another measure. Finally, as shown in the
decision matrix, the type of hotspot also conditions the suitability of the RRM preliminarily selection. The sets of RRM
obtained according to the decision matrix for each of the determinants are merged, and finally the most restricted
recommendation is considered.

| RRM Code | Risk Reduction Measure | Topography | Geology | | Land cover | | | Types of HS | | | | Prioritization Stakeholders ranking |
| | | | | | | | | Conservation | | | | |
| | | Flood prone areas (Low-lying/wadis) | Bare non-consolidated | Bare consolidated | Built-up | Crop land | Covered by vegetation | Lagoons/ mangroves | Turtle nesting areas | Touristic areas | Relevant infrastructures | |
|---|---|---|---|---|---|---|---|---|---|---|---|---|
| PR. 1 | Social and Institutional Raising awareness | ++ | + | + | ++ | + | + | ++ | ++ | ++ | ++ | 1 |
| EM. 1 | Emergency Planning Early Warning Systems | ++ | + | + | ++ | + | + | + | + | ++ | ++ | 2 |
| PR. 3 | Social and Institutional Education | ++ | + | + | ++ | + | + | ++ | ++ | ++ | ++ | 3 |
| RA. 1 | Hazard, Vulnerability and Risk Assessment | ++ | ++ | ++ | ++ | ++ | ++ | ++ | ++ | ++ | ++ | 4 |
| EM. 2 | Emergency Planning Evacuation planning | ++ | + | + | ++ | ++ | + | + | + | ++ | ++ | 5 |
| PR. 2 | Social and Institutional Capacity building | ++ | + | + | ++ | + | + | ++ | ++ | ++ | ++ | 6 |
| EN. 2 | Breakwaters | ++ | + | + | ++ | + | + | - | - | + | ++ | 7 |
| NA. 3 | Artificial sand dunes and dune restoration | ++ | ++ | + | - | + | ++ | - | ++ | + | + | 8 |
| NA. 4 | Living shorelines | ++ | + | - | ++ | + | ++ | ++ | ++ | + | + | 9 |
| PL. 3 | Coastal setbacks | ++ | + | + | ++ | + | ++ | + | + | + | + | 10 |
| NA. 5 | Wetland restoration | ++ | + | - | - | + | ++ | ++ | ++ | + | + | 11 |
| PL. 1 | Building standards | ++ | + | + | ++ | + | + | + | + | + | + | 12 |
| EN. 4 | Land claim | ++ | + | + | + | + | + | - | - | + | ++ | 13 |
| NA. 2 | Beach nourishment | ++ | ++ | + | - | + | ++ | - | ++ | + | + | 14 |
| PL. 2 | Flood proofing | ++ | + | + | ++ | + | + | + | + | + | + | 15 |
| NA. 1 | Managed realignment | ++ | + | - | - | + | + | - | ++ | + | + | 16 |
| EN. 1 | Seawalls and sea dykes | ++ | + | + | ++ | + | + | - | - | + | ++ | 17 |
| EN. 3 | Movable barriers and closure dams | ++ | + | + | ++ | + | + | - | - | + | ++ | 18 |

**Table 3. Decision matrix for the selection of recommended RRM (++: highly recommended; +: recommended; -: not recommended).**
**Last column: prioritization of RRM according to key stakeholder ratings on Oman risk management. The matrix is presented**
**ordered by these prioritization results.**
Finally, in the third phase, the prioritization analysis considers the characteristics of each measure, its technical and economic
requirements, efficiency and durability, the SWOT analysis and the capacity of the country to implement them. In addition to
technical criteria, there are subjective aspects, including local knowledge and expertise, which should be taken into account
when selecting certain recommended RRM as preferred over others. Results of this preferences, shown in figure Figure 7, are
also reflected in the sorting of Table 3, based on the last column.



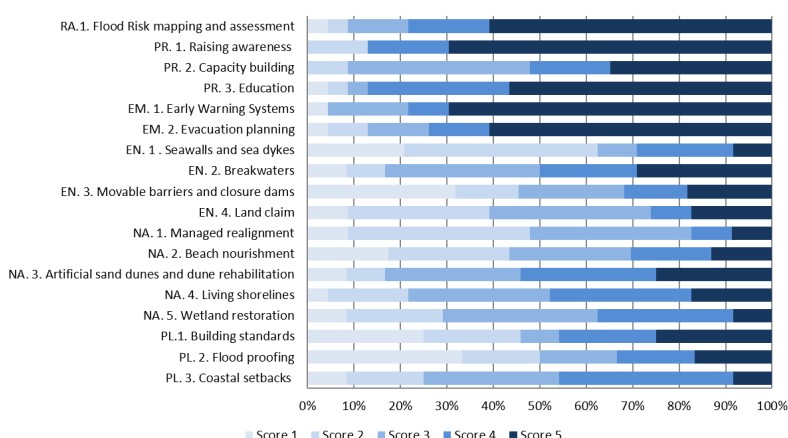

**Figure 7. Scoring of the RRM according to the stakeholder panel ratings (1: the least preferred; 5: most preferred)**

## 3 Results

This section presents two types of results. First, sections 3.1 Tsunami risk assessmentand 3.2 Tsunami risk reduction in Omandeal with technical results obtained from the application of the methodology to the Sultanate of Oman. They describe the most relevant results of the tsunami risk assessment and one example regarding the risk reduction measures selected and prioritized for a specific site. Finally, section 3.3 Science-based support for the tsunami DRR decision making processdescribe the management tools developed and its usefulness for the tsunami DRR decision-making process.

### 3.1 Tsunami risk assessment

The tsunami hazard analysis indicates that the greater flooded area is located in the northern plain and in one section of the eastern face of the country, as shown in figure Figure 8a (country's wilayats are sorted from north to south in this and following graphs). However, the greater the flooded area does not imply necessarily the greater the impact. In fact, the vulnerability analysis show that the elements at risk are not homogenously distributed along these flooded areas. The greater values for the exposure are on the northern plain, especially between Shinas and Bawshar Wilayats (see figure Figure 8b and Figure 8c). Saham, Suwayq, Al Musanaah, Barka and As Seeb Wilayats have the highest percentage of exposed population, all above 10%, the latter two more than 15%, whereas there is almost no exposure in the coastline from Sur to Dalkut Wilayats, with most of relative values below 1%. The Wilayat Al Jazir, even if having a low absolute number of exposed population, it represents about the 8% of the total, ranking on the side of the most exposed in relative terms. Regarding the exposure of buildings and infrastructures, the pattern is very similar. Higher rates of exposure take place in the northern area, especially from Sinas to As Seeb Wilayats (with exposure values over 40%), with the exception of Liwa. In the rest of the country Jaalan Bani Bu Ali and Al Jazir have the highest values, with 45% (about 8,300 items) and 25% (about 750 elements) respectively. The vulnerability assessment reveal the different characteristics of each wilayat in terms of both population and infrastructure, being the highest values correlated to the highest exposure values. In general, the most representative variables of the human vulnerability assessment along the entire coast are the "expatriates" and the "sensitive age groups", both around the 30% of the total population exposed (Figure 8b). The variable that contributes less to the human vulnerability is "disable", but even if not very representative in relative values (about 2% of total exposure), it was maintained in the analysis because of its relevance and importance within the risk assessment.





As for the infrastructure dimension (Figure 8c and Figure 8d), the vulnerability analysis highlights that "critical buildings"
category are the most affected, being around 96% of all sensitive and exposed buildings. The 70% of the buildings within this
class are religious, being the wilayats Saham and As Suwayq the most affected. Despite their lower absolute number, it is
necessary to consider the other variables that feed the infrastructure vulnerability analysis due to their significant relevance in
case of an emergency (emergency, supply, dangerous and strategic), as described in the risk assessment section. In this sense,
Figure 8d shows their distribution along the coastal wilayats, highlighting Sohar, where ten petrochemical industries, three
container terminals, two bulk liquid terminals, one general cargo terminal and a sugar refinery could be affected. All of these
industries are located within the area and surroundings of the Port of Sohar.

Integrated vulnerability results are shown in Figure 9a for both human and infrastructure dimensions. According to the
vulnerability classification, the colour ramp varies from green to red, being the green the lowest value of the index and red the
highest. Note that, for a better understanding, the representation is at the wilayat level, while the vulnerability analysis is
performed exclusively for the potentially inundated area due to the tsunami hazard considered. The highest vulnerability scores
mainly corresponds with the wilayats located in the northern plain area. Analysing the differences among them, it may be
concluded that the most vulnerable wilayats (sorted from north to south) are Sohar, Saham (highest IVI score), As Suwayq,
Barka, As Seeb (highest HVI score) and Bawshar.





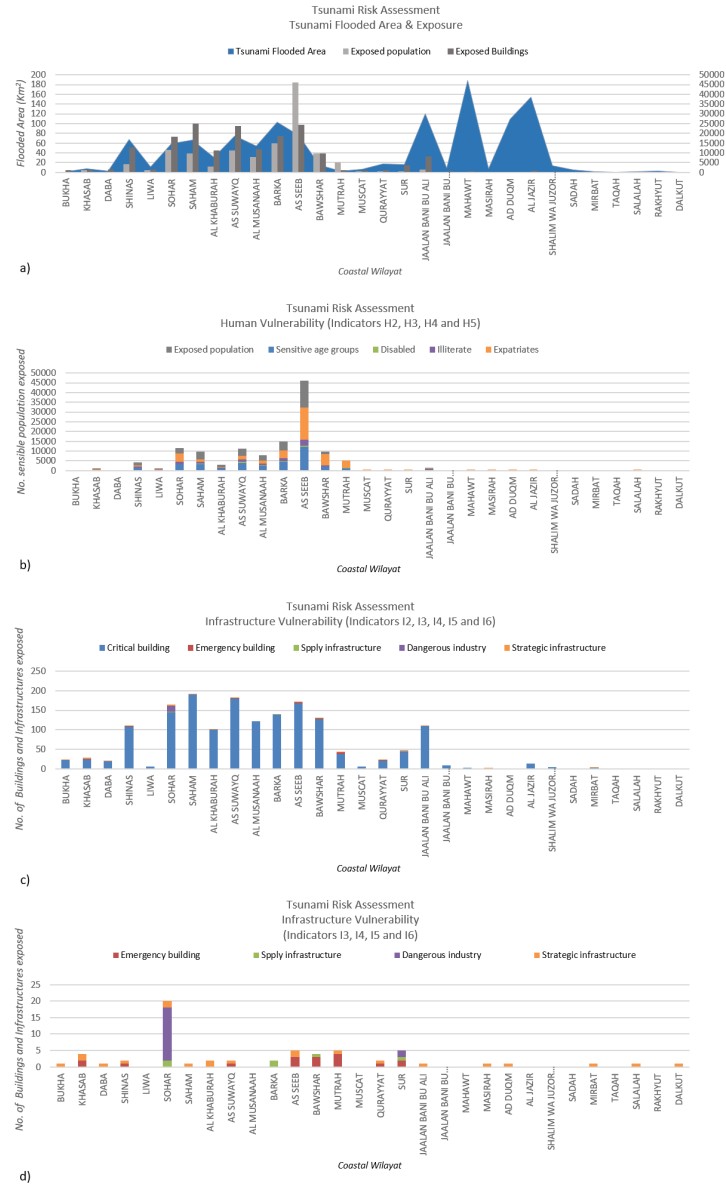


**Figure 8. Tsunami Risk assessment: (a) Tsunami flooded area and exposure, (b) Human exposure and vulnerability variables, (c) and (d) Infrastructures exposure and vulnerability variables.**

Finally, Figure 9b shows the integrated risk map as a synthesis, indicating the amount of area disaggregated by each risk level
and wilayat, which permits to know the amount of population and infrastructures per level. Therefore, it is shown that the
northern area of the country would be the most affected by the tsunami scenario modelled in this work, both because of the
greater impact of the hazard and the higher degree of exposure and vulnerability.






**Figure 9. (a) IVI and HVI: Infrastructure and human vulnerability indexes; (b) Integrated tsunami risk assessment**



Summarizing tsunami risk results, Figure 10a shows the distribution of the exposed population by risk level and wilayat, the
greater consequences being on As Seeb and Barka wilayats. Almost 55% of the exposed population is located in very high-
risk areas and around 25% in high-risk areas. Regarding the infrastructure dimension, most of the exposed built-up area is
located in medium risk zones (about 60%), and around a 25% in high-risk zones. Less than 1% of the built up area result in
very high infrastructure risk areas. Built-up area by risk level and wilayat is presented in Figure 10b, showing that Sohar and
As Seeb are the most affected wilayats both in terms of built-up area exposure and risk level.

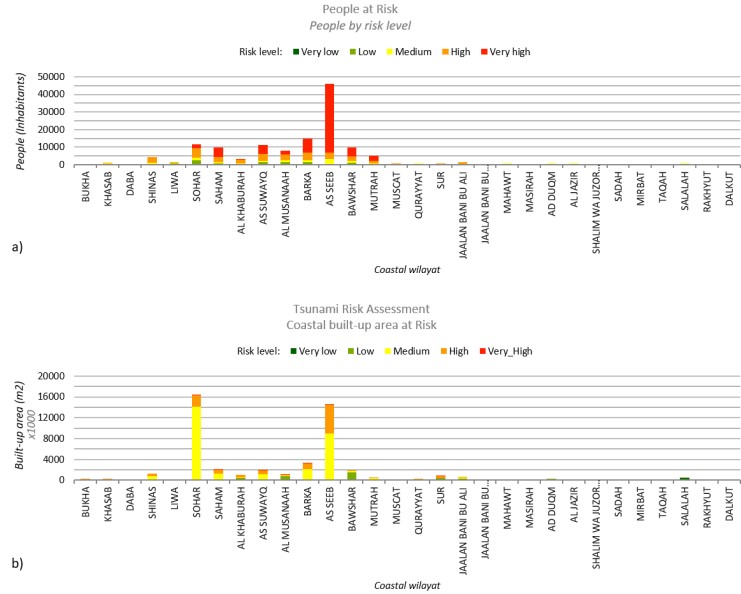


**Figure 10. People and built up area by risk level**
**3.2 Tsunami risk reduction in Oman**
The methodology applied for the selection and prioritization of optimal RRM, resulted in the identification of 89 hot spots
(HS) along the entire coast of the country, half of them located on the north coast, mainly from Liwa to Sur wilayats. About
25% of them are concentrated in the southeast area of the country, especially in wilayats Salalah (12) and Sadah (9). Mashira
and Ad Duqm concentrates 10 and 5 HS respectively. According to the method followed, 79 out of the initial 89 were assigned
with a set of RRM.
Next, an example is included to show the whole procedure, focused on the wilayat As Seeb. This wilayat concentrates the
largest amount of population exposed to the highest level of risk and is the second wilayat with the greatest infrastructures risk
level. The target area (the HS) is the Muscat International Airport and surroundings where, in addition to the airport itself, is
located the building of the Public Authority for Civil Aviation of Oman (PACA) that houses the Multi Hazard Early Warning
System and the National Tsunami Warning Centre.
Figure 11 shows the selected HS, a simple view of the risk assessment results, a summary of the characterization, and the
preliminary set of RRM recommended resulting from the decision matrix. The list is shorted (most preferred on top) according
to the prioritization made by the stakeholder panel, based on their knowledge and expertise on the feasibility and the
institutional, economic and technological capacity of the country for their implementation.





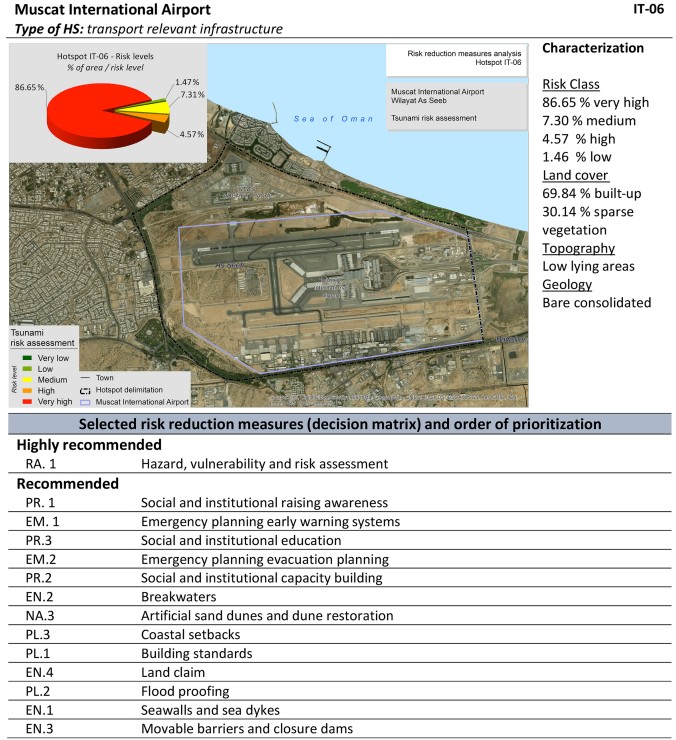

| Selected risk reduction measures (decision matrix) and order of prioritization | |
|---|---|
| **Highly recommended** | |
| RA. 1 | Hazard, vulnerability and risk assessment |
| **Recommended** | |
| PR. 1 | Social and institutional raising awareness |
| EM. 1 | Emergency planning early warning systems |
| PR.3 | Social and institutional education |
| EM.2 | Emergency planning evacuation planning |
| PR.2 | Social and institutional capacity building |
| EN.2 | Breakwaters |
| NA.3 | Artificial sand dunes and dune restoration |
| PL.3 | Coastal setbacks |
| PL.1 | Building standards |
| EN.4 | Land claim |
| PL.2 | Flood proofing |
| EN.1 | Seawalls and sea dykes |
| EN.3 | Movable barriers and closure dams |


**Figure 11. RRM preliminary proposal for Wilayat As Seeb relevant infrastructure area**

The first six recommended RRM are related to the preparedness strategy. Based on this result, the implementation of these measures require specific supplementary studies at a greater resolution. These may be: high-resolution data collection for the risk analysis (topo-bathymetry, tsunamigenic sources characterization, and vulnerability), in-depth numerical modelling of the flooding physical process, development of a strategy for education of critical groups (most vulnerable members, leaders, institutions, government, educators, etc.), and the cooperation between the government, relief agencies and local communities to enhance the early warning systems and the evacuation planning process.

Regarding the prevention strategy, the first recommended countermeasure is the construction of breakwaters (EN. 2 in Figure 11). Tsunami breakwaters are usually constructed in the mouth of a bay or estuary, not in open coasts. However, according to the general workflow developed and presented in Figure 1 (point 6) a detached breakwater has been modelled to understand the efficiency of the measure. The model resulted in a local increase in wave elevation in the study area (see Figure 12 Figure 12b and Figure 12c). Therefore, although more detailed studies would be necessary, this prevention measure should be discarded at this site. The second recommended prevention measure is the "artificial sand dunes and dune restoration". Accordingly, a more detailed study has been done in a subset of the area by means of modelling an artificial sand dune with a crest height of 3 metres, showing an efficient reduction of the flooded area, as shown in Figure 12d.




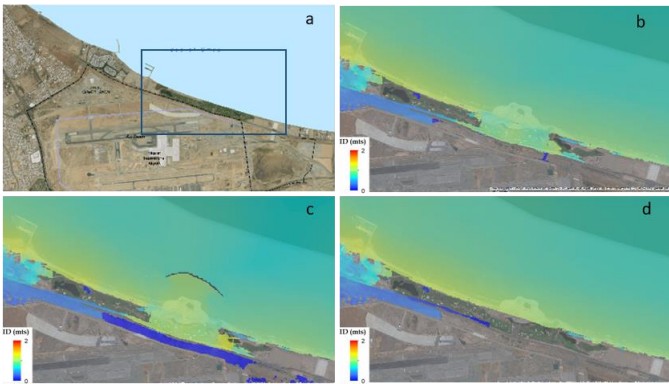


**Figure 12 . Detailed analysis of preliminary engineering RRM: a) Zoomed sample area; b) Modelled flooded area; c) with the breakwater option; d) with artificial sand dune option.**

Similar procedures for obtaining a preliminary set of RRM have been developed for all the hotspots and for some local areas.
In-depth studies should be made to perform a second stage analysis of the recommended countermeasures, considering higher
resolution of the hazard analysis and detailed information provided by the vulnerability variables and indicators.

**3.3 Science-based support for the tsunami DRR decision making process**

One of the main objectives of the study is to improve tsunami risk management through the effective use of the results
obtained. In this sense, science and technical results are translated into two risk management tools: (i) the Tsunami Hazard,
Vulnerability and Risk Atlas, and (ii) the Risk Reduction Measures Handbook. These tools have been implemented and
activated by the Directorate General of Meteorology of Oman (DGMET). In addition, a knowledge and technology transfer
strategy has been carried out to ensure adequate long-term management.
The "Tsunami Hazard, Vulnerability and Risk Atlas", contains a comprehensive description of the methodology applied to
assess the risk and all maps from the hazard analysis and vulnerability variables and indices to the final risk results. It is
expected to be used as the main source for awareness and education regarding tsunamis and as the basis for further local and
detailed studies. In this regards, DGMET efforts are focused in distributing and conducting follow-up meetings to all
involved stakeholders, including Supreme Council for Planning, Ministry of Education, The Public Authority Of Radio And
Television, National Committee for Civil Defence (NCCD), Public Authority for Civil Defence and Ambulance and Royal
Oman Police-Operation. Follow up meetings are also included in the general strategy to explain the atlas information and
discuss the best approaches to utilize such information for the planning and implementing policies and strategies.
The "Tsunami Risk Reduction Measures Handbook" is a useful manual to help in the decision-making process related with
the tsunami prevention and preparedness. It includes a brief explanation of the methodology developed to select and
recommend each set of measures, the catalogue of RRM, containing individual RRM-cards for each countermeasure and the
results obtained for each area along the coast of Oman, including the set of recommended RRM for each specific location.
Similar to the hazard, vulnerability and risk atlas, DGMET has forwarded the handbook to the government cabinet to
distribute among all stakeholders, especially to the Supreme Council for Planning.
Finally, as an additional result of this study, a web based tool to support the tsunami early warning system (called MHRAS)
was also developed, implemented and linked to the DGMET Decision Support System.



## 4 Conclusions

Integrated risk assessments are essential for identifying the most vulnerable communities and worst expected consequences, as well as for designing and planning a roadmap towards risk reduction. For this reason, they should be the basis to link scientific and technical advances with appropriate decision-making and effective risk management.

The methodology presented was developed to build an effective connection between tsunami risk assessment and tsunami risk reduction, with the objective of supporting risk managers by facilitating science-based decision-making in the phases of prevention and preparedness, before an event occurs.

The tsunami hazard modelling, based on potential earthquake sources, permitted to perform an analysis to identify the worst possible scenario, considering the low frequency/high severity nature of the hazard. Thus, it permitted to estimate the worst negative consequences as the main outcome of the risk assessment. The potentially most affected areas in Oman, in terms of tsunami-prone flooded areas are the northern plain of the country especially Barka and As Seeb as well as Mahawt and Al Jazir wilayats on the eastern area.

The semi quantitative indicator-based approach for the vulnerability and risk assessment, which integrates risk components (hazard, exposure and vulnerability) and the human and infrastructure dimensions, has been proved useful to discern the more sensitive areas from a human-centred perspective. The indicators system is helpful for the decision-making process in two ways. First, the information at the index and indicator level allows a broad insight of where the exposed elements are and which are more susceptible to suffering the impact of the hazard, i.e., where to focus the efforts towards risk reduction. Second, the approach permits to easily track back to the variables. This information is essential to understand the precise root causes of vulnerability and risk results, to be tackled by adequate and specific DRR measures. In Oman, the most vulnerable areas are located in the northern plain of Oman, highlighting wilayat As Seeb, both in the human and infrastructure dimension and wilayats Saham and Suwayq in the infrastructure dimension. The eastern part, although affected by the inundation, is not so vulnerable. The combination of hazard and vulnerability assessments reveals that the worst expected consequences are for As Seeb and Barka wilayats in terms of human risk and for Sohar and As Seeb in terms of infrastructure risk, according to the tsunami modelled in this work.

As for the connection between risk assessment results and risk management, for each defined tsunami-risk management area, the methodology allows identifying, selecting and prioritizing, a preliminary set of suitable and site-specific RRM. This analysis discards non-suitable measures and allows a more in-depth exploration, defining the basis for analysing the feasibility of its implementation, including its technical and economic viability. Through the example shown for the area of Muscat International Airport, it has been illustrated the usefulness of the methodology, which can be applied in other parts of the world facing other natural events that may trigger a disaster.

In this sense, with the aim of producing a useful outcome for the risk management, all the results obtained and the detailed description of the method were compiled in two handy management tools. These tools permit to analyse and facilitate the decision-making, to replicate and to update the study by the tsunami disaster managers of Oman, thus contributing to the connection between science-based risk results and disaster risk management.

The involvement and support of relevant stakeholders in charge of the risk management process is essential for the success and usefulness of the method. Their encouragement has been one of the priorities throughout the application of the method in the case study with the aim of achieving the main objective of minimizing the consequences that a potential tsunami could trigger. In this sense, the work aims to contribute to the implementation of the Sendai Framework, by understanding the current tsunami risk, by the commitment of stakeholders and by the linkage between risk outcomes and risk reduction measures.





**5 Acknowledgements**
The authors thank the Ministry of Transport and Communications of the Government of the Sultanate of Oman (MOTC), the
Public Authority for Civil Aviation (PACA) and the Directorate General of Meteorology (DGMET), for supporting and
funding the project "*Assessment of Coastal Hazards, Vulnerability and Risk for the Coast of Oman" during the period 2014-*
*2016*. We also thank and appreciate the collaboration of the International Oceanographic Commission of the United Nations
Educational, Scientific and Cultural Organization personnel (IOC-UNESCO).

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
