# Peer review of "From Tsunami Risk Assessment to Disaster Risk Reduction. The case of Oman"

_Natural Hazards and Earth System Sciences, 2017_

## Referee Comment (RC1) · Anonymous Referee #1 · 22 Jan 2018

Main comments - How can you evaluate that your goal was succeed for tsunami DRR even if there is no actual tsunami event to test? Of course, I agreed if your goal is to develop some tools or frameworks for DRR and to say that the country will be more prepared. Otherwise, please give some examples (may be in other countries?) to support that in what way, what you have achieved in this project can reduce tsunami risk in the future. - Risk communication is also very important. Good quality of DRR countermeasures will be meaningless if they were failed in transferring to people at risk. Also, I could see that you mentioned about education, but I think it should be explained more on how the people at risk will be properly/correctly educated and have high capacity enough to receive risk information from the government, etc.

Specific comments Title: I feel that the title is rather general and should be modified to

be more attractive Abstract: I feel that the main results of your study did not appear in the abstract. I would also write about the recommended countermeasures, recommendation for DRR in Oman here. Introduction: - You may split this part into three sections, 1) tsunami hazards in Oman, 2) risk assessment method and 3) your study objectives - These are other studies on tsunamis in MSZ and should be properly credited. I remember that one of them also use high resolution of bathymetry in Oman. Heidarzadeh M, Kijko A (2011) A probabilistic tsunami hazard assessment for the Makran subduction zone at the northwestern Indian Ocean. Nat Hazards 56:577–593 Heidarzadeh M, Satake K (2014a) New insights into the source of the Makran tsunami of 27 November 1945 from tsunami waveforms and coastal deformation data. Pure Appl Geophys 172(3):621–640 Heidarzadeh M, Satake K (2014b) Possible sources of the tsunami observed in the northwestern Indian Ocean following the 2013 September 24 Mw 7.7 Pakistan inland earthquake. Geophys J Int 199(2):752–766 Heidarzadeh M, Pirooz MD, Zaker NH, Synolakis CE (2008a) Evaluating tsunami hazard in the Northwestern Indian Ocean. Pure appl Geophys 165:2045–2058 Heidarzadeh M, Pirooz MD, Zaker NH, Yalciner AC, Mokhtari M, Esmaeily A (2008b) Historical tsunami in the Makran Subduction Zone off the southern coasts of Iran and Pakistan and results of numerical modeling. Ocean Eng 165:2045–2058 Heidarzadeh M, Pirooz MD, Zaker NH (2009) Modeling the near-field effects of the worst-case tsunami in the Makran subduction zone. Ocean Eng 36(5):368–376 Latcharote, P., Al-Salem, K., Suppasri, A., Pokavanich, T., Toda, S., Jayaramu, Y., Al-Enezi, A., Al-Ragumand, A. and Imamura, F. (2017) Tsunami hazard evaluation for Kuwait and Arabian Gulf due to Makran Subduction Zone and Subaerial landslides, Natural Hazards - Page 2 lines 40-43: This way of citing is not so good. Because you are mentioning three different risk targets (building, infrastructure and human), readers will not know that which reference did what. - There are recent studies on the vulnerability of the mentioned risk targets (in addition to building). Suppasri, A., Fukui, K., Yamashita, K., Leelawat, N., Ohira, H., and Imamura, F.: Developing fragility functions for aquaculture rafts and eelgrass in the case of the 2011 Great East Japan tsunami, Nat. Hazards Earth Syst. Sci.,

18, 145-155 Shoji, G. and Nakamura, T.: Damage assessment of road bridges subjected to the 2011 Tohoku Pacific earthquake tsunami, Journal of Disaster Research, 12, 79–89, 2017. Suppasri, A., Latcharote, P., Bricker, J. D., Leelawat, N., Hayashi, A., Yamashita, K., Makinoshima, F., Roeber, V. and Imamura, F. (2016) Improvement of tsunami countermeasures based on lessons from the 2011 great east japan earthquake and tsunami -Situation after five years-, Coastal Engineering Journal, 58 (4), 1640011. Suppasri, A., Muhari, A., Futami, T., Imamura, F. and Shuto, N. (2014) Loss functions of small marine vessels based on surveyed data and numerical simulation of the 2011 Great East Japan tsunami, Journal of Waterway, Port, Coastal and Ocean Engineering-ASCE, 140 (5), 04014018. Methodology - You may write section name 2.1, 2.2., 2.3. . . in Fig. 1. 2.1: Please give a reference that other source of tsunamis such as landslide or volcanic eruption can be neglected. - Page 5 line 129: "Okada model" should be properly cited giving the year and put in the reference - Please also tell readers about your computational grid size. Although the simulation was done by your previous study but the grid size is important to understand the resolution of your study. - Please give some comments if the tsunami sources in your study the same or different to other previous studies. - Page 5 line 145: "drag level" sounds wired to me. I would prefer "drag force" or "hydrodynamic force". Please check and consider. 2.2: I feel that you just mentioned about your risk variables but not on how the hazard and risk will be linked. Few sentences in lines 146-150 is probably rather fit to this section as they explain the linkage between hazard and vulnerability. However, another question about these references is how can you directly applied their proposed vulnerability functions to Oman. For example, building strength in Oman may different to other countries. - Table 1: I think age and gender are also important as they are directly related to the evacuation speed. Did you used different kinds of vulnerability functions for different kinds of buildings/infrastructures? 2.3 Fig. 4: I can see that you used flow depth and drag force as your hazard index. What if both give different results? Low flow depth with high velocity will have high drag force, therefore, you will have lower hazard level when using flow depth but higher hazard level when using drag

force. - What is the meaning of "assigned score", how it is assigned and how it was applied to different human and infrastructure index? - There should be some explanations about the hazard-vulnerability table, not just only shown in Fig. 4. 2.4: What is RRM? - Fig. 5: "exposure assessment" have never mentioned before or in any places in your paper but shown in this figure. Please explain in your main text. - In Fig. 2 you show disaster cycle, but you only focused on prevention and preparation in your study. How emergency response and recovery included in your study or will be considered in the future? - I can see only section 2.4.1 but no 2.4.2. - Page 10 line 278: How the recommended measures were determined? In what way they were decided that priority to be recommend? Were they determined by hazard reduction performance, economic cost, B/C, impact to environment, etc? Results: Fig. 9: How can local people get an access to information like in Fig. 9? - 3.3 Page 18 Lines 395-396: How the knowledge can be transferred? Any example? - Page 18 Line 405: How can you make sure that it will not be just a manual which people will never read? How this manual will be used for various practical actions such as evacuation drills, etc? Page 18 Line 411: In what way the warning message can be disseminated to local people or how they can access? Conclusions: I suggest reorganizing like this 1) the new method used in this study, 2) recommendations to government or local people in Oman and 3) Global applications/limitations of this study - The Sendai Framework have never appeared in the main text but suddenly mentioned here. If you want to keep this sentence, please also mention in your introduction or methodology on the linkage between your work and the Sendai Framework.

---

## Referee Comment (RC2) · Anonymous Referee #2 · 4 May 2018

**General Comments**

The paper by Aguirre-Ayerbe et al. deals with tsunami risk assessment and strategies for risk reduction along the coast of Oman, presenting a comprehensive and integrated approach, that starts from the scientific aspect (hazard assessment), includes engineering methodologies (such as vulnerability indicators), and involves also the operative and human dimensions (involvement of stakeholders). Another important aspect is for sure the study and quantification of the human dimension of vulnerability, usually neglected or ignored in tsunami vulnerability and risk assessment. The approach adopted in this work, bridging different aspects and methodologies, is gaining importance in natural risk reduction perspective.

In general, the manuscript is clear, well organized and well written (some remarks are reported below, in the "Technical Corrections" section). The results provide very interesting indications to the local authorities in terms of tsunami hazard and effectiveness of preparedness and preventions measures. The references are extensive and appropriate, such as no particular remarks are found concerning the pictures.

The methodology section, on the contrary, needs some improvements (reported in "Specific Comments" section), probably leaving too much descriptions and details to other related works, where a similar approach or part of it was applied.

The main weakness of the work is that no observations on tsunami hazard and vulnerability are reported, in order to understand if the proposed vulnerability indicators fit the local conditions (for example, building vulnerability classes are the same in Oman as the case considered in SCHEMA project?), and if the proposed countermeasures are really effective. In few words: it is possible to validate in some way all the assumptions taken for all the aspects (hazard, vulnerability and exposure, risk, countermeasures) considered.

Apart from this aspect, the paper represents an important step forward the integration between scientific and operational aspects, and is recommended for publication with minor revisions.

**Specific Comments**

1. INTRODUCTION

Has this approach been applied to other cases? Which difficulties and could raise in other areas, and which changes should you perform on the vulnerability indicators?

2. METHODOLOGY

Line 130. Maybe it is better to specify that COMCOT account also for land flooding using the moving boundary technique.

Lines 135-137. When dealing with deterministic hazard assessment is this area, are there non-seismic tsunami events that are worth of consideration? Landslide-tsunamis, for example, usually affects short coastal stretches but their effect can be highly destructive.

At Line 140, an early warning system establishment for Oman is cited. Is it working, in phase of realization, or just an intention at the moment?

In the paper the expression "inundation depth" is used repeatedly (for example in the definition of the drag level, Line 145): if it refers to the height of the water inundating the land (meaning the difference between the elevation of the water top and the topography) it is better to use the expression "flow depth".

When dealing with building vulnerability assessment, the most diffused quantities in tsunami science are flow depth $H$, water velocity $V$, and momentum flux (defined as $HV^2$), the last accounting for the energy of

the incoming wave. However, specify better in Lines 145-150 that drag force is for human dimension and flow depth for building one, and justify why you did not use momentum flux.

What do you mean with "exposed people and infrastructures" (Table 1 and Line 172)? Are they counted considering their inclusion in the flooded area? Explain and specify better.

Concerning Risk Assessment: how are hazard components for human and building components estimated? Is the flow depth over each building computed as the maximum water height? And what about drag force? Is it computed at each time step and then the maximum selected, or is it simply the product of maximum flow depth and velocity for each element? Consider that these do not occur necessarily at the same time.

Lines 198-199: how are the two risk dimensions weighted?

Line 266: again about exposure, here concerning HS. How is it measured? Is there a threshold for the flow depth, or is it sufficient the inclusion in the flooded area in order to consider the element "exposed"? Specify and clear better this point.

3. RESULTS

Line 302. When you speak of "flooded area" do you consider a flow depth threshold? Or is it sufficient that the area is simply covered by water, though few centimeters?

Lines 376-383. Can you provide some explanation of the fact that a detached breakwater would increase wave elevation on the coast? Are there some hydrodynamics effects justifying it? In Figures 12c and 12d probably it would be better to evidence where such prevention measures (breakwater and artificial dunes) have been placed.

**Technical Corrections**

ABSTRACT

the first sentence [Lines 9 to 11] is repeated almost exactly in the Introduction [Lines 24-25], change one of the two.

1. INTRODUCTION

Line 32. "most exposed to MSZ *effects*"

Line 39. "for all the components *contributing to* the risk"

Line 48. "have to be taken" instead of "are to take"

Line 72. Remove comma after "or"

2. METHODOLOGY

Line 125. "to" instead of "and"

Line 127. Remove "quake", repetition with Line 126.

Line 128. "…COMCOT (Wang, 2009), which *solves shallow water equations using* Okada model…"

Line 129. Provide citation for Okada model

Lines 162 and 164. Remove comma after "are"

Line 185. Describe in few words (or include a reference about) the min-max method.

Line 253. "It is summarize*d*"

Line 279. "On  one hand"

Line 280. "where flooding occurs on a regular basis, at least annually" this seems to mean that these areas are affected by tsunami at least once per year. Is "flooding" meant in general, by storms, river flooding or other? Reformulate better.

3. RESULTS

Line 296. Separate with space "assessmentand"

Line 297. Separate with space "Omandeal"

Line 299. Separate with space "processdescribe"

Line 309. Remove "it", the subject Wilayat Al Jazir (is already present)

Line 310. Remove "the" before "8%".

Lines 361-362. Move "is located" at the end of the sentence.

Line 423. Add comma after "tsunami-prone flooded areas".

Line 438. Remove comma after "prioritizing".

---

## Author Comment (AC1) · 15 Jun 2018

Aguirre-Ayerbe, Ignacio – Manuscript nhess-2017-448

RESPONSES TO REFEREE #1

Dear reviewer,

First, we really appreciate Referee's #1 valuable comments and suggestions, which offer us an opportunity to improve the paper. We found all comments and additional references provided very interesting and believe that consequent changes in the paper represent an improvement over the initial submission. Below you will find your comments followed by our response. We have also attached a new version of the manuscript (Aguirre-Ayerbe_From TRA to DRR_Discussion_Manuscript_v2) with the changes proposed after your suggestions, marked in green. In addition, you will also find the changes anticipated following the suggestions of a second reviewer, which are highlighted in blue. Lines referred in this author's response are the lines numbered in the version 2 of the manuscript, which is attached to this response.

General comments Â■ GENERAL COMMENTS: How can you evaluate that your goal was succeed for tsunami DRR even if there is no actual tsunami event to test? Of course, I agreed if your goal is to develop some tools or frameworks for DRR and to say that the country will be more prepared. Otherwise, please give some examples (may be in other countries?) to support that in what way, what you have achieved in this project can reduce tsunami risk in the future. Risk communication is also very important. Good quality of DRR countermeasures will be meaningless if they were failed in transferring to people at risk. Also, I could see that you mentioned about education, but I think it should be explained more on how the people at risk will be properly/correctly educated and have high capacity enough to receive risk information from the government, etc.

GENERAL RESPONSE: We agree with your general comment, the goal of this study is to develop and provide a framework and some tools to improve the preparedness of the country to a tsunami event. The tsunami risk assessment performed, together with the risk reduction measures identified are essential for the risk-management preparedness strategy. Thus, improving preparedness will rise the capacity of the country in facing a tsunami event.

We also agree with your comment regarding risk communication and education. Risk communication and education helps to raise awareness and consequently improve effectiveness of certain measures. Performing a tsunami hazard, vulnerability and risk atlas as well as risk reduction measures handbook is a pillar for communication processes. Risk assessment and mapping is indeed the first step in the risk management process. Without these tools is not possible to have proper knowledge of the potential problem. These tools were developed to be included in the tsunami risk management process, but actual implementation into government policies and institutional commu- nication strategy or educational official curricula goes beyond the scope of this work. It is precisely the next step from this study. This idea has been included in current lines 427-428. Specific comments 1. REVIEWER COMMENT: Title: I feel that the title is rather general and should be modified to be more attractive. RESPONSE: We agree with the reviewer in the idea that the title is rather general. However, the objective that led us to define this title was an attempt to synthetize what is presented as much as possible through the main keywords, i.e. tsunami/risk assessment/disaster risk reduc- tion/in Oman, so that anyone interested in the topic and in the topic in Oman will easily find this article. We prefer and suggest keeping the current title.

2. REVIEWER COMMENT: Abstract: I feel that the main results of your study did not appear in the abstract. I would also write about the recommended countermeasures, recommendation for DRR in Oman here. RESPONSE: We totally agree. Following this recommendation, a paragraph has been included highlighting main results (please, see current lines 19-22).

3. REVIEWER COMMENT: Introduction: You may split this part into three sections: 1) tsunami hazards in Oman, 2) risk assessment method and 3) your study objectives. RESPONSE: We agree with the structure proposed by the reviewer, which is indeed the structure followed. We have tried to divide in the proposed sections but they fragment too much the introduction, which is not so long (just one side) to integrate subdivisions, so we decided to redo and suggest the initial proposal.

4. REVIEWER COMMENT: Introduction: These are other studies on tsunamis in MSZ and should be properly credited. I remember that one of them also use high resolu- tion of bathymetry in Oman. Heidarzadeh M, Kijko A (2011) A probabilistic tsunami hazard assessment for the Makran subduction zone at the northwestern Indian Ocean. Nat Hazards 56:577–593. Heidarzadeh M, Satake K (2014a) New insights into the source of the Makran tsunami of 27 November 1945 from tsunami waveforms and coastal deformation data. Pure Appl Geophys 172(3):621–640 Heidarzadeh M, Satake K (2014b) Possible sources of the tsunami observed in the northwestern Indian Ocean following the 2013 September 24 Mw 7.7 Pakistan inland earthquake. Geophys J Int 199(2):752–766 Heidarzadeh M, Pirooz MD, Zaker NH, Synolakis CE (2008a) Evaluating tsunami hazard in the Northwestern Indian Ocean. Pure appl Geophys 165:2045–2058 Heidarzadeh M, Pirooz MD, Zaker NH, Yalciner AC, Mokhtari M, Esmaeily A (2008b) Historical tsunami in the Makran Subduction Zone off the southern coasts of Iran and Pakistan and results of numerical modeling. Ocean Eng 165:2045–2058 Heidarzadeh M, Pirooz MD, Zaker NH (2009) Modeling the near-field effects of the worst-case tsunami in the Makran subduction zone. Ocean Eng 36(5):368–376 Latcharote, P., Al-Salem, K., Suppasri, A., Pokavanich, T., Toda, S., Jayaramu, Y., Al-Enezi, A., Al-Ragumand, A. and Imamura, F. (2017) Tsunami hazard evaluation for Kuwait and Arabian Gulf due to Makran Subduction Zone and Subaerial landslides, Natural Hazards. RESPONSE: We appreciate the reviewer information regarding additional references. One of them was already cited. The rest of them have been included (please, see green coloured highlights in section 1 Introduction).

5. REVIEWER COMMENT Page 2 lines 40-43: This way of citing is not so good. Because you are mentioning three different risk targets (building, infrastructure and human), readers will not know that which reference did what. - There are recent studies on the vulnerability of the mentioned risk targets (in addition to building). Suppasri, A., Fukui, K., Yamashita, K., Leelawat, N., Ohira, H., and Imamura, F.: Developing fragility functions for aquaculture rafts and eelgrass in the case of the 2011 Great East Japan tsunami, Nat. Hazards Earth Syst. Sci., 18, 145-155. Shoji, G. and Nakamura, T.: Damage assessment of road bridges subjected to the 2011 Tohoku Pacific earthquake tsunami, Journal of Disaster Research, 12, 79–89, 2017. Suppasri, A., Latcharote, P., Bricker, J. D., Leelawat, N., Hayashi, A., Yamashita, K., Makinoshima, F., Roeber, V. and Imamura, F. (2016) Improvement of tsunami countermeasures based on lessons from the 2011 great east japan earthquake and tsunami -Situation after five years-, Coastal Engineering Journal, 58 (4), 1640011. Suppasri, A., Muhari, A., Futami, T., Imamura, F. and Shuto, N. (2014) Loss functions of small marine vessels based on surveyed data and numerical simulation of the 2011 Great East Japan tsunami, Journal of Waterway, Port, Coastal and Ocean Engineering-ASCE, 140 (5), 04014018. RESPONSE: We agree with this reviewer's comment. The initial idea was simply to highlight the difference between different approaches from a wider point of view, but we think the reviewer comment is appropriate and have made changes following this suggestion. We have also included two of the references suggested in this comment (Suppasri et al., 2018 and Shoji and Nakamura 2017). Please see all proposed changes along the current lines 45 to 50.

6. REVIEWER COMMENT: Methodology: You may write section name 2.1, 2.2., 2.3. . . in Fig. 1. RESPONSE: Numbers in figure 1 represent the different orderly steps in which the disaster risk reduction is carried out in this study. However, sections of the document do not follow exactly the same numbering (for example, exposure and vulnerability are treated and explained together). Therefore, we believe that writing section numbers in the figure would be confusing since they would be different from the current ordinal numbers (1 to 6). As the reviewer can verify, explanation of the figure including the corresponding sections is already included in current lines 78-95.

7. REVIEWER COMMENT: 2.1: Please give a reference that other source of tsunamis such as landslide or volcanic eruption can be neglected. RESPONSE: In this study, we have just considered potential earthquake sources for the tsunami risk assessment. We cannot neglect other source for tsunami generation in the area. We have slightly modify the sentence in current line 127 to make it clearer that in this study we have considered only earthquake sources.

8. REVIEWER COMMENT: Page 5 line 129: "Okada model" should be properly cited giving the year and put in the reference - Please also tell readers about your computational grid size. Although the simulation was done by your previous study but the grid size is important to understand the resolution of your study. - Please give some comments if the tsunami sources in your study the same or different to other previous studies. RESPONSE: We thank the reviewer for this comment. We forgot to include

Okada reference. This has been now included in current line 135 and 593. Regarding resolution, it is included in current line 137.

Tsunami sources are original from our study. Just one is based on previous studies (Heidarzadeh et al., 2008) and included in the manuscript in the current line 148. There is another scientific article detailing this process in preparation and will be submitted this month. 9. REVIEWER COMMENT: Page 5 line 145: "drag level" sounds wired to me. I would prefer "drag force" or "hydrodynamic force". Please check and consider. RESPONSE: According to the reviewer comment, we have made a clarification in current line 152-153, referring to the term also as "depth-velocity product" as it is called in the reference considered (Jonkman 2008), which is a proxy of the drag force to which the reviewer refers to. We have also maintained the concept "drag level" as it is used in previous works, e.g. González-Riancho et al. (2014). Please, see current lines where changes have been made: 152-153 and 212-213.

10. REVIEWER COMMENT: 2.2: I feel that you just mentioned about your risk variables but not on how the hazard and risk will be linked. Few sentences in lines 146-150 is probably rather fit to this section as they explain the linkage between hazard and vulnerability. RESPONSE: After a brief mention in the "methodology" section (current lines 89-93), the main explanation on how hazard and vulnerability are combined may be found in "risk assessment" section (current lines 203-217). We thought it was useful to also include some lines under the "hazard assessment" section, to explain that hazard variables are classified (current lines 154-158). Besides, following the reviewer comment, we have added a clarification (current line 154-155).

11. REVIEWER COMMENT: Table 1: I think age and gender are also important as they are directly related to the evacuation speed. How can you directly applied their proposed vulnerability functions to Oman. For example, building strength in Oman may different to other countries. Did you used different kinds of vulnerability functions for different kinds of buildings/infrastructures? RESPONSE: As information on building materials were not available, we considered as a minimum, based on field observa-

tions, that buildings included within the infrastructure dimension fit at least with class C1 of Tinti and Valencia references (Brick with reinforced column & masonry filling. One or two storeys), so we used the corresponding damage function.

12. REVIEWER COMMENT: 2.3 Fig. 4:I can see that you used flow depth and drag force as your hazard index. What if both give different results? Low flow depth with high velocity will have high drag force, therefore, you will have lower hazard level when using flow depth but higher hazard level when using drag force. RESPONSE: Yes, that is true. However, the analysis is independent for each dimension. We used drag level (as a proxy of drag force) for the human dimension (based on previous works, among them Jonkman et al., 2008). On the other hand, we used flow depth for the infrastructure dimension (based among others in the work developed by Tinti 2001 and Valencia 2011). Afterwards, we combine each hazard variable level with vulnerability level (for each dimension) to obtain human and infrastructure risk indexes respectively. Both risk indexes can be combined later to obtain an aggregated risk index, thanks to the indicators and indexes system applied. This is explained mainly in lines 212-214.

13. REVIEWER COMMENT: What is the meaning of "assigned score", how it is assigned and how it was applied to different human and infrastructure index? There should be some explanations about the hazard-vulnerability table, not just only shown in Fig. 4. RESPONSE: "Assigned score" in figure 4 refers to the vulnerability classification, which is mainly described in the "vulnerability assessment" section (current lines 194-197). The classification of the hazard index is described in the "hazard assessment" section (current lines 154-159). Following the reviewer comment and to avoid confusion we have changed Fig.4 to follow exactly the same terminology: instead of "assigned score", it now says "vulnerability class".

14. REVIEWER COMMENT:2.4: What is RRM? RESPONSE: RRM is the acronym for "Risk Reduction Measures". It appears for the first time on page 1 and since then the acronym is used since the term appears 30 times.

15. REVIEWER COMMENT: Fig. 5: "exposure assessment" have never mentioned before or in any places in your paper but shown in this figure. Please explain in your main text. RESPONSE: Exposure is one of the risk components, as explained in current line 68. It is also stated that exposure is a necessary component (as they are the hazard and vulnerability) for the establishment of risk reduction strategies and measures (current lines 68-70 and 87-93). By exposure assessment, we are referring to the analysis of people, buildings and infrastructures located in a flooded area as described in current lines 88-89 and in the modified ones 181-182. We have also cited, in brackets, the risk components "(hazard, exposure and vulnerability)" (current line 224-225) to improve understanding.

16. REVIEWER COMMENT: In Fig. 2: you show disaster cycle, but you only focused on prevention and preparation in your study. How emergency response and recovery included in your study or will be considered in the future? RESPONSE: Reviewer comment is right. In this study, we have proposed a framework for the whole disaster risk management cycle but focused only on pre-event strategies, prevention and preparedness (please, see current lines 103-105). Post event measures should be considered in the future. Nonetheless, it must be considered that each of the strategies includes several actions that may overlap in time and that may even belong to more than one strategy. In this sense, there are some preparedness measures, which are oriented to the post-event phase of the disaster management, such as contingency planning, stockpiling of equipment and supplies and arrangement for coordination.

17. REVIEWER COMMENT: I can see only section 2.4.1 but no 2.4.2. RESPONSE: Yes. We have included 2.4.1, 2.4.2 and 2.4.3 sections. Please see current lines 269, 285 and 300.

18. REVIEWER COMMENT: Page 10 line 278: How the recommended measures were determined? In what way they were decided that priority to be recommend? Were they determined by hazard reduction performance, economic cost, B/C, impact to environment, etc.? RESPONSE: Risk reduction measures were determined based on the technical information described in the RRM-cards (RRM-cards are described in current lines 251 to 257; we have also included a sentence in current lines 290-291 to clarify) and depend on the site-specific conditions that have determined the type hotspot (hotspot determination is explained in current lines 269-284).

19. REVIEWER COMMENT: Results: Fig. 9: How can local people get an access to information like in Fig. 9? RESPONSE: All the information generated in this study have been included in the "Tsunami Hazard, Vulnerability and Risk Atlas" and the "Risk Reduction Measures Handbook". This information have been transferred to the Government of Oman and it is expected to be used as the main source for policy planning, awareness and education regarding tsunami disaster.

20. REVIEWER COMMENT: 3.3 Page 18 Lines 395-396: How the knowledge can be transferred? Any example? RESPONSE: The knowledge was transferred to government authorities and technicians by means of technical courses on tsunami hazard, tsunami vulnerability and risk, GIS for disaster risk reduction and system procedures and architecture. This capacity building ensure a long-term management of the product developed (as mentioned in current line 410-411). Please, see following links: http://www.ioc-tsunami.org/index.php?option=com_content&view=article&id=269:assessment-of-coastal-hazards-vulnerability-and-risk-for-the-coast-of-oman&catid=20&lang=en&Itemid=68

http://www.unesco.org/new/en/member-states/single-view/news/oman_launches_an_early_warning_system_to_address_

21. REVIEWER COMMENT: Page 18 Line 405: How can you make sure that it will not be just a manual which people will never read? How this manual will be used for various practical actions such as evacuation drills, etc? RESPONSE: This study is the necessary starting point for the reviewer commented actions. Several copies of this manual were delivered to government authorities. Several follow-up meeting were held with different stakeholders to explain the information and discuss the best approaches to utilize such information for the planning and implementing policies and strategies. The manual is also expected to be used as the main source for public awareness and educational purposes. The long term follow-up is out of the scope of the work presented.

22. REVIEWER COMMENT: Page 18 Line 411: In what way the warning message can be disseminated to local people or how they can access? RESPONSE: That issue is out of the scope of the presented study. The tsunami early warning system is only accessible for tsunami risk authorities/managers (i.e., DGMET) and they are the responsible to define the emergency protocol.

23. REVIEWER COMMENT: I suggest reorganizing like this 1) the new method used in this study, 2) recommendations to government or local people in Oman and 3) Global applications/limitations of this study. RESPONSE: We thank the reviewer suggestion to reorganize the conclusions section. The structure of this section follows each of the steps (methodology) explained in the paper, in the same order that they are initially presented. Section 3.3 has been maintained under the "results" section since it refers to the outcomes of the study and their usefulness for tsunami risk management in the country. Following reviewer suggestion, we have changed the last paragraph (about stakeholders involvement) leaving the paragraphs about usefulness and overall application of the methodology and brief description of outcomes and their usefulness at the end (please, see current lines 462-469).

24. REVIEWER COMMENT: The Sendai Framework have never appeared in the main text but suddenly mentioned here. If you want to keep this sentence, please also mention in your introduction or methodology on the linkage between your work and the Sendai Framework. RESPONSE: We thank the reviewer's comment. In fact, this sentence was initially linked to a part of the introduction that was discarded and we forgot to delete it in the conclusions. Sendai framework sentence in the "conclusions" sections has been deleted.

Please also note the supplement to this comment:
https://www.nat-hazards-earth-syst-sci-discuss.net/nhess-2017-448/nhess-2017-448-AC1-supplement.pdf

**Supplement:**

[revised manuscript text omitted]

---

## Author Comment (AC2) · 15 Jun 2018

Aguirre-Ayerbe, Ignacio – Manuscript nhess-2017-448

RESPONSES TO REFEREE #2

First of all we really thank the Referee#2 for accepting the revision of the paper and for the opportunity offered to improve it through the valuable comments and suggestions proposed. We also appreciate a lot the technical revision and the corrections proposed. It is a great contribution for the improvement of the initial submission.

Below you will find your comments followed by our response. We have also attached a new version of the manuscript (Aguirre-Ayerbe_From TRA to DRR_Discussion_Manuscript_v2) with the changes proposed after your suggestions,

marked in blue. In addition, you will also find the changes anticipated following the suggestions of a second reviewer, which are highlighted in green. Lines referred in this author's response are the lines numbered in the version 2 of the manuscript attached to this response.

General Comments

The paper by Aguirre-Ayerbe et al. deals with tsunami risk assessment and strategies for risk reduction along the coast of Oman, presenting a comprehensive and integrated approach, that starts from the scientific aspect (hazard assessment), includes engineering methodologies (such as vulnerability indicators), and involves also the operative and human dimensions (involvement of stakeholders). Another important aspect is for sure the study and quantification of the human dimension of vulnerability, usually neglected or ignored in tsunami vulnerability and risk assessment. The approach adopted in this work, bridging different aspects and methodologies, is gaining importance in natural risk reduction perspective. In general, the manuscript is clear, well organized and well written (some remarks are reported below, in the "Technical Corrections" section). The results provide very interesting indications to the local authorities in terms of tsunami hazard and effectiveness of preparedness and preventions measures. The references are extensive and appropriate, such as no particular remarks are found concerning the pictures. The methodology section, on the contrary, needs some improvements (reported in "Specific Comments" section), probably leaving too much descriptions and details to other related works, where a similar approach or part of it was applied. The main weakness of the work is that no observations on tsunami hazard and vulnerability are reported, in order to understand if the proposed vulnerability indicators fit the local conditions (for example, building vulnerability classes are the same in Oman as the case considered in SCHEMA project?), and if the proposed countermeasures are really effective. In few words: it is possible to validate in some way all the assumptions taken for all the aspects (hazard, vulnerability and exposure, risk, countermeasures) considered. Apart from this aspect, the paper represents an important step forward the integration between scientific and operational aspects, and is recommended for publication with minor revisions.

General response:

We thank the reviewer the analysis and reflections on this study. Major past tsunamis in Oman are not very detailed documented in terms of physical and human impacts, so no historical references are available to "calibrate" or "validate" the assessment . For the hazard assessment, one of the scenario considered is the historical event of 1945 (Heidarzadeh net al., 2008). For the vulnerability and exposure, present conditions have to be analysed (unless the objective would be to compare with past situations, which is not the case). For the building vulnerability function applied, it has been selected from SCHEMA study, based on similar building characteristics in Oman (these data come from post-tsunami observations collected by several authors in Indonesia in the aftermath of the 2004 Indian Ocean tsunami). Regarding the effectiveness of the measures, each measure included in the set of RRM proposed is based on previous studies (UNFCC, 1999; Nicholls et al., 2007; UNESCO, 2009a, Linham et al., 2010) and analysed and characterised by considering technical and economic requirements, possible supplementary measures, efficiency, durability and initial cost analysis. Besides, local (country) capacities to implement them is analysed based on the information provided by the ad-hoc (local) experts group panel. In addition, a SWOT analysis has been performed for each measure, in which experts and past experiences are considered. Each measure (developed on RRM-cards format, as pointed out in the paper) incorporates a bibliographic reference list.

In conclusion, local characteristics and other experiences have been considered as much as possible. This said, is important to clarify that the goal of this study is to provide a framework and some management tools to improve the preparedness of the country to a tsunami event. The tsunami risk assessment performed, together with the risk reduction measures identified are essential for the risk-management preparedness strategy. Thus, improving preparedness will improve the capacity of the country to face a tsunami event.

Specific Comments

**1. INTRODUCTION**

REVIEWER COMMENT: Has this approach been applied to other cases? Which difficulties and could raise in other areas, and which changes should you perform on the vulnerability indicators?

RESPONSE: There are several studies and international DRR institutions applying indicators-based approaches to perform risk assessments to several hazards, some of them mentioned in current lines 54-60. These studies are very helpful to carry out an appropriate selection and definition of the indicators, at different temporal and spatial scales. Some of them have been validated considering past events (e.g. World Risk Index; González-Riancho et al., 2015; Papathoma-Khole, 2016). Following these works, some basic indicators (analytically and statistically sound) should not be ever neglected. If we consider an assessment with a similar scope and scale of work, local conditions should be considered as much as possible in the definition of the indicators, for the integration of context-specific problems. These local characteristics are usually related to very detailed information and limitations often appear regarding data availability and/or quality and confidence. This is one of the main constrains/limitations. Indicators in general must be appropriate in scope, understandable, easy to interpret and comparable. Some clarifications following these ideas have been included in lines 464-465.

2. METHODOLOGY REVIEWER COMMENT: Line 130. Maybe it is better to specify that COMCOT account also for land flooding using the moving boundary technique.

RESPONSE: We agree with this reviewer's comment and have included this idea in current line 135-136.

REVIEWER COMMENT : Lines 135-137. When dealing with deterministic hazard assessment is this area, are there non-seismic tsunami events that are worth of consideration? Landslide-tsunamis, for example, usually affects short coastal stretches but their effect can be highly destructive.

RESPONSE: There are other possible sources of generation, as evidenced in previous studies (for example, Heidarzadeh and Satake, 2014a and 2014b and 2017; Suppasri et al., 20106). However, in this study, we have just considered potential earthquake sources for the tsunami risk assessment. Landslides, as mentioned by the reviewer and some of the references cited, have a local effect (even if highly destructive) and the efforts and resources needed to analyse them for the entire country go beyond the scope of this study. We have slightly modify the sentence in current line 127 to make it clearer that in this study we have considered only earthquake sources.

REVIEWER COMMENT : At Line 140, an early warning system establishment for Oman is cited. Is it working, in phase of realization, or just an intention at the moment?

RESPONSE: The early warning system is currently working. We have slightly modified the sentence in current line 146-147 to specify it. Please, see also the link provided for additional information: http://www.unesco.org/new/en/media-services/single-view/news/oman_launches_an_early_warning_system_to_address_natural_dis/ http://www.helzel.com/files/432/upload/Pressreleases/2015/NMHEWS-Oman-2015.pdf

REVIEWER COMMENT: In the paper the expression "inundation depth" is used repeatedly (for example in the definition of the drag level, Line 145): if it refers to the height of the water inundating the land (meaning the difference between the elevation of the water top and the topography) it is better to use the expression "flow depth".

RESPONSE: We have changed the expression "inundation depth" to "flow depth" along the document (and figure 4). We have also referred to it as inundation depth when is first described (current line 152), since there are some works that already call it that

way. Please, see changes in current lines 87, 151, 157 and 214 and Fig.4.

REVIEWER COMMENT: When dealing with building vulnerability assessment, the most diffused quantities in tsunami science are flow depth H, water velocity V, and momentum flux (defined as HV2), the last accounting for the energy of the incoming wave. However, specify better in Lines 145-150 that drag force is for human dimension and flow depth for building one, and justify why you did not use momentum flux.

RESPONSE: As it is properly expressed by the reviewer, different tsunami hazard variables may be applied to assess building vulnerability. In the case of the present study, we based on the works developed by Tinti (2011) and Valencia (2011) where the flow depth-building damage relationship is analysed to develop fragility curves, based on post-tsunami observations that consider different building typologies (structure, construction material, number of storeys). This is explained in current lines 157-159 and 214. The use of flow depth variable for infrastructure dimension and depth-velocity product (drag level) for the human dimension is explained in the "risk assessment" section, current lines 204-218.

REVIEWER COMMENT: What do you mean with "exposed people and infrastructures" (Table 1 and Line 172)? Are they counted considering their inclusion in the flooded area? Explain and specify better.

RESPONSE: By exposed people and infrastructures, we are referring to the people, buildings and infrastructures located in a flooded area, as described in current lines 88-89. A sentence has been included in current lines 181-182 explaining better the exposure.

REVIEWER COMMENT: Concerning Risk Assessment: how are hazard components for human and building components estimated? Is the flow depth over each building computed as the maximum water height? And what about drag force? Is it computed at each time step and then the maximum selected, or is it simply the product of maximum flow depth and velocity for each element? Consider that these do not occur necessarily at the same time.

RESPONSE: Yes, you are right and this is a very good question. Hazard variables are calculated at each time step and the maximum is then selected: h(max) or (h*u)max. There is another scientific article detailing this process in preparation and will be submitted this month. A brief explanation has been included in current line 149.

REVIEWER COMMENT: Lines 198-199: how are the two risk dimensions weighted?

RESPONSE: The whole analysis is performed through a human-centred perspective. In this sense, a slightly higher weight has been considered for the human dimension.

REVIEWER COMMENT: Line 266: again about exposure, here concerning HS. How is it measured? Is there a threshold for the flow depth, or is it sufficient the inclusion in the flooded area in order to consider the element "exposed"? Specify and clear better this point. RESPONSE: It is sufficient the inclusion in the flooded area. No threshold has been established. We have included a clarification in current line 277.

3. RESULTS

REVIEWER COMMENT: Line 302. When you speak of "flooded area", do you consider a flow depth threshold? Or is it sufficient that the area is simply covered by water, though few centimetres?

RESPONSE: No threshold has been established.

REVIEWER COMMENT: Lines 376-383. Can you provide some explanation of the fact that a detached breakwater would increase wave elevation on the coast? Are there some hydrodynamics effects justifying it? In Figures 12c and 12d probably it would be better to evidence where such prevention measures (breakwater and artificial dunes) have been placed.

RESPONSE: The presence of a break water modifies tsunami height and energy flow direction, generating an accumulation of energy in the leeside, focusing the affection to the coast and increasing the flooded area. A brief explanation has been included in current lines 393-394. Figures 12c and 12d include now location of breakwater and potential artificial dune location.

Technical Corrections

ABSTRACT

REVIEWER COMMENT: the first sentence [Lines 9 to 11] is repeated almost exactly in the Introduction [Lines 24-25], change one of the two. 1. INTRODUCTION REVIEWER COMMENT: Line 32. "most exposed to MSZ effects"

RESPONSE: Done, please see current line 37.

REVIEWER COMMENT: Line 39. "for all the components contributing to the risk"

RESPONSE: Done, please see current line 44.

REVIEWER COMMENT: Line 48. "have to be taken" instead of "are to take"

RESPONSE: Done, please see current line 55.

REVIEWER COMMENT: Line 72. Remove comma after "or"

RESPONSE: Done, please see current line 79.

2. METHODOLOGY

REVIEWER COMMENT: Line 125. "to" instead of "and"

RESPONSE: Done, please see current line 131.

REVIEWER COMMENT: Line 127. Remove "quake", repetition with Line 126.

RESPONSE: Done, please see current line 132.

REVIEWER COMMENT: Line 128. "...COMCOT (Wang, 2009), which solves shallow water equations using Okada model..."

RESPONSE: Done, please see current line 135.

REVIEWER COMMENT:Line 129. Provide citation for Okada model

RESPONSE: Done, please see current line 135.

REVIEWER COMMENT: Lines 162 and 164. Remove comma after "are"

RESPONSE: Done, please see current line 171 and 173.

REVIEWER COMMENT: Line 185. Describe in few words (or include a reference about) the min-max method.

RESPONSE: please see current line 195 and 592.

REVIEWER COMMENT: Line 253. "It is summarized"

RESPONSE: Done, please see current line 263.

REVIEWER COMMENT: Line 279. "On the one hand"

RESPONSE: Done, please see current line 291.

REVIEWER COMMENT: Line 280. "where flooding occurs on a regular basis, at least annually" this seems to mean that these areas are affected by tsunami at least once per year. Is "flooding" meant in general, by storms, river flooding or other? Reformulate better.

RESPONSE: Done, please see current line 292.

3. RESULTS

REVIEWER COMMENT: Line 296. Separate with space "assessmentand"

RESPONSE: Done, please see current line 309

REVIEWER COMMENT: Line 297. Separate with space "Omandeal"

RESPONSE: Done, please see current line 309.

REVIEWER COMMENT: Line 299. Separate with space "processdescribe"

RESPONSE: Done, please see current line 312.

REVIEWER COMMENT: Line 309. Remove "it", the subject Wilayat Al Jazir (is already present)

RESPONSE: Done, please see current line 322.

REVIEWER COMMENT: Line 310. Remove "the" before "8%".

RESPONSE: Done, please see current line 323.

REVIEWER COMMENT: Lines 361-362. Move "is located" at the end of the sentence.

RESPONSE: Done, please see current line 376.

REVIEWER COMMENT: Line 423. Add comma after "tsunami-prone flooded areas".

RESPONSE: Done, please see current line 441

REVIEWER COMMENT: Line 438. Remove comma after "prioritizing".

RESPONSE: Done, please see current line 456.

Please also note the supplement to this comment:
https://www.nat-hazards-earth-syst-sci-discuss.net/nhess-2017-448/nhess-2017-448-AC2-supplement.pdf

**Supplement:**

[revised manuscript text omitted]

---

## Editor Comment (EC1) · U. Ulbrich (Editor) · 20 Aug 2018

Editorial note to
Nat. Hazards Earth Syst. Sci. Discuss., nhess-2017-448, 2018
https://doi.org/10.5194/nhess-2017-448-editorial-note

[Figure]

[Figure]

Natural Hazards and Earth System Sciences

**Editorial note to**
**"From tsunami risk assessment to disaster risk reduction – the case of Oman", Nat. Hazards Earth Syst. Sci. Discuss., nhess-2017-448, 2018**

**Uwe Ulbrich**[1,*]

[1]Institute for Meteorology, Freie Universität Berlin, 12165 Berlin, Germany
[*]Executive editor

**Correspondence:** Uwe Ulbrich (ulbrich@met.fu-berlin.de)

Published: 20 August 2018

Please note that one figure was exchanged in the discussion paper following a formal request, without any changes to the content. There is no change in the figure caption, nor in the content of the article, just in the image of one figure.

**Published by Copernicus Publications on behalf of the European Geosciences Union.**

---

## Author Response (AR2)

Aguirre-Ayerbe, Ignacio – Manuscript nhess-2017-448

Iteration: Correction

Response to Editor

Dear Dr. Didenkulova,

Thank you for the revision of our manuscript entitled: *From Tsunami Risk Assessment to Disaster Risk Reduction. The case of Oman*.

In this document you can find the answer to the two comments of Referee #2. At the end of the document, you will find the re-edited manuscript with changes highlighted in green.

Ignacio Aguirre Ayerbe

*Corresponding author*

**Editor Decision: Publish subject to technical corrections** (10 Jul 2018) by Ira Didenkulova
Non-public comments to the Author:
Please, respond to the following two comments of the referee:

**1- REVIEWER COMMENT:** Section 2.3: I still believe that force is more important than flow depth in terms of damage to infrastructure. If the authors used flow depth for infrastructure (based on previous studies), I guess it is not about the damage to structure but the "function of infrastructure". Higher flow depth might interrupt longer period of infrastructure? I feel more acceptable in this way of explanation. Please consider if a bit more clarification is necessary.

**1- RESPONSE:** Thank you for this comment; it was useful to include an additional clarification. We agree that, as stated by the reviewer, force is a detailed approach to assess damage to infrastructures and is a research area of great interest. However, there are also other ways to assess damage to infrastructures. In our case, as explained in the response to the Referee #1 question 5 (regarding comments to point 2-Methodology), we based on previous works developed by Tinti (2011) and Valencia (2011) where the flow depth/building damage relationship is analysed to develop fragility curves, based on post-tsunami observations that consider different building typologies (structure, construction material, number of storeys), flow depth and damage analysis. This is explained in current lines 157-159 and 214.

We have also included an additional explanation in current lines 214-216.

**2- REVIEWER COMMENT**: Page 2 Line47, Sohi --> Shoji
**2- RESPONSE**: Thank you very much for this correction. Change was done (please, see current line 47)

[revised manuscript text omitted]

**PREPAREDNESS**
- RISK ASSESSMENT AND MAPPING
- SOCIAL AND INSTITUTIONAL EDUCATION /CAPACITY BUILDING
- EMERGENCY PLANNING
  - EVACUATION PLANNING
  - EARLY WARNING SYSTEMS
  - CONTINGENCY PLANNING
  - STOCKPILING OF EQUIPMENT & SUPPLIES
  - ARRANGEMENTS FOR COORDINATION)

**PREVENTION**
- ENGINEERING BASED
- NATURE-BASED
- COASTAL PLANNING, LAND USE REGULATION AND ARCHITECTURAL.

**TSUNAMI**

**RESEARCH**
- SOCIAL / ECONOMIC
- GEOLOGY
- ENGINEERING
- ECOLOGY
- RISK ANALYSIS
- EVENT ANALYSIS
- INSURANCE
- POLICY DEVELOPMENT

**EMERGENCY**
- ALERT / INSTRUCTIONS
- EVACUATION ACTION
- EMERGENCY ACTION
- RESCUE
- SHORT TERM RELIEF

**RECOVERY**
- RECONSTRUCTION
- REHABILITATION (SUPPLY AND DISPOSALS, COMMUNICATION, TRANSPORT)
- DAMAGE ANALYSIS AND COMPENSATION

PREPAREDNESS · EMERGENCY / RESPONSE · PRE-EVENT · POST-EVENT · PREVENTION · RECOVERY

[revised manuscript text omitted]

**1** Determination of hotspots (HS)

Identification of hotspots, HS

- Relevant infrastructures
- Touristic areas
- Conservation areas (S)

Exposed — No

Yes

Risk Class ≥ Medium — No — •Conservation •Touristic

No

Yes — Yes

Characterization of selected HS

- Topography
- Geology
- Land Cover

**2** Selection of Risk Reduction Measures

Decision Matrix (S)

**3** Prioritization of Risk Reduction Measures

- Technical capacity
- Financial capacity
- Knowledge
- Preference (S)

**Figure 6. Scheme of the methodology for the prioritization of recommended tsunami risk reduction measures (S: participation of**
**stakeholder panel of local and international experts on coastal and risk management).**

**2.4.1 Determination of hotspots**

The first step is the determination of hotspots, which are the zones in which RRM will be further proposed. Coastal hotspots (HS) are identified in consensus with the stakeholder panel, including built-up populated areas and the following areas of special interest: (i) relevant infrastructures such as transport and communications infrastructures (airports and sea-ports), supply infrastructures (power and water) and dangerous infrastructures (refineries, dangerous industries areas and military bases); (ii) touristic regions, where there is significant seasonal variation in the population and (iii) environmental conservation areas, to consider the fragile and complex systems where the coastal ecosystems converge with the marine dynamics and the human activities, which include lagoons, mangroves and turtle nesting areas.

After the identification of the HS, it is evaluated whether they are exposed to tsunami hazard (i.e. located in the flooded area)

and if they exceed the risk class threshold as shown in **Figure 6**, in order to determine the units that will feed the decision matrix into the second phase. Because of their significance, the scarcity of data when performing the vulnerability assessment and the relevance given by local stakeholders, touristic regions and environmental conservation areas will move to the next step if the HS is exposed, regardless the risk level. In all other cases, for those HS under very low, low risk or not expose, no countermeasures will be assigned. The HS characterization is carried out by assigning elevation characteristics (highlighting low-lying areas and wadis), a geology categorization (bare consolidated or non-consolidated substratum) and the land cover (cropland, built-up areas and vegetation-covered areas).

**2.4.2 Selection of risk reduction measures**

The second stage consists in the preliminary assignment of RRM to each HS according to the decision matrix. The matrix, which was validated by the stakeholder panel, is fed by the specific characteristics of each HS and by type of HS, as described previously. **Table 3** shows the decision matrix, already sorted by the ratings of the stakeholder panel of experts on coastal risk management in Oman, as explained in section 2.4.3.

The assignment of each recommended measure (highly recommended, recommended or not recommended) is based on the information described in each of the RRM-cards and depends on the characteristics that have determined the type HS. On one hand, the topography of the area, with a focus on the low-lying areas and wadis, where coastal and pluvial flooding occurs on a regular basis, at least annually. Likewise, the geology and land cover is analysed to consider the bedrock and type of land use, that condition the suitability of one or another measure. Finally, as shown in the decision matrix, the type of hotspot also conditions the suitability of the RRM preliminarily selection. The sets of RRM obtained according to the decision matrix for each of the determinants are merged, and finally the most restricted recommendation is considered.

[revised manuscript text omitted]